# Chemical crystallography by serial femtosecond X-ray diffraction

Elyse A. Schriber[1,2,15], Daniel W. Paley[3,15], Robert Bolotovsky[3], Daniel J. Rosenberg[3,4], Raymond G. Sierra[5], Andrew Aquila[5], Derek Mendez[3], Frédéric Poitevin[5], Johannes P. Blaschke[6], Asmit Bhowmick[3], Ryan P. Kelly[1,2], Mark Hunter[5], Brandon Hayes[5], Derek C. Popple[6,7], Matthew Yeung[8], Carina Pareja-Rivera[9], Stella Lisova[10], Kensuke Tono[11], Michihiro Sugahara[12], Shigeki Owada[11], Tevye Kuykendall[13], Kaiyuan Yao[14], P. James Schuck[14], Diego Solis-Ibarra[9], Nicholas K. Sauter[3✉], Aaron S. Brewster[3✉] & J. Nathan Hohman[1,2✉]

Inorganic–organic hybrid materials represent a large share of newly reported structures, owing to their simple synthetic routes and customizable properties[1]. This proliferation has led to a characterization bottleneck: many hybrid materials are obligate microcrystals with low symmetry and severe radiation sensitivity, interfering with the standard techniques of single-crystal X-ray diffraction[2,3] and electron microdiffraction[4–11]. Here we demonstrate small-molecule serial femtosecond X-ray crystallography (smSFX) for the determination of material crystal structures from microcrystals. We subjected microcrystalline suspensions to X-ray free-electron laser radiation[12,13] and obtained thousands of randomly oriented diffraction patterns. We determined unit cells by aggregating spot-finding results into high-resolution powder diffractograms. After indexing the sparse serial patterns by a graph theory approach[14], the resulting datasets can be solved and refined using standard tools for single-crystal diffraction data[15–17]. We describe the ab initio structure solutions of mithrene (AgSePh)[18–20], thiorene (AgSPh) and tethrene (AgTePh), of which the latter two were previously unknown structures. In thiorene, we identify a geometric change in the silver–silver bonding network that is linked to its divergent optoelectronic properties[20]. We demonstrate that smSFX can be applied as a general technique for structure determination of beam-sensitive microcrystalline materials at near-ambient temperature and pressure.

Single-crystal X-ray diffraction (SCXRD) is a foundational characterization technique for chemistry and materials science, with over one million published organic and metal–organic structures[2,3]. Frequently encountered challenges for SCXRD include difficulty in crystal growth, instability to atmosphere, solvent loss and radiation sensitivity. Powder diffraction and electron microdiffraction are established methods for determining structures of microcrystalline substances, but each can be limited by some combination of the challenges above. As an example, the structure of the excitonic and blue-emitting mithrene, $AgSeC_6H_5$, was determined by SCXRD[18–20] but the remaining and optically divergent silver benzenechalcogenolates ($AgEC_6H_5$, E = S, Te) have not yielded to characterization by Rietveld refinement or micro-electron diffraction[21]. In this work, we present the room-temperature (298 K) crystal structures of all three materials as determined by X-ray free-electron laser (XFEL) small-molecule serial femtosecond crystallography (smSFX). We first

validate the method against the known structure of mithrene before presenting the previously unknown structures of the homologues thiorene, $AgSC_6H_5$, and tethrene, $AgTeC_6H_5$. Although the Ag–E coordination is similar in all three systems, we find a motif of linear Ag–Ag interactions in thiorene that provides insight into its lack of optical emission compared to the trigonal planar Ag–Ag network in the other two. smSFX is a general method with which to structurally characterize microcrystalline, low-symmetry and radiation-sensitive materials.

Electron diffraction and powder X-ray diffraction (PXRD) are currently the dominant techniques for structural studies on microcrystals. In PXRD work, peak broadening arises from sample-based and instrument-based factors and complicates the search for a unit cell and the measurement of structure factors. Therefore, Rietveld refinement is often more suitable for known structures or those with known homologues[22,23]. When unknowns are characterized, the

[1]Institute of Materials Science, University of Connecticut, Storrs, CT, USA. [2]Department of Chemistry, University of Connecticut, Storrs, CT, USA. [3]Molecular Biophysics and Integrated Bioimaging Division, Lawrence Berkeley National Laboratory, Berkeley, CA, USA. [4]Graduate Group in Biophysics, University of California, Berkeley, CA, USA. [5]Linac Coherent Light Source, SLAC National Accelerator Laboratory, Menlo Park, CA, USA. [6]National Energy Research Scientific Computing Center, Lawrence Berkeley National Laboratory, Berkeley, CA, USA. [7]College of Chemistry, University of California, Berkeley, Berkeley, CA, USA. [8]Department of Electrical Engineering and Computer Science, Massachusetts Institute of Technology, Cambridge, MA, USA. [9]Instituto de Investigaciones en Materiales, Universidad Nacional Autónoma de México, Coyoacán, Mexico. [10]Department of Physics, Arizona State University, Tempe, AZ, USA. [11]SPring-8, Japan Synchrotron Radiation Research Institute, Sayo, Japan. [12]RIKEN SPring-8 Center, Sayo, Japan. [13]The Molecular Foundry, Lawrence Berkeley National Laboratory, Berkeley, CA, USA. [14]Department of Mechanical Engineering, Columbia University, New York, NY, USA. [15]These authors contributed equally: Elyse A. Schriber, Daniel W. Paley. ✉e-mail: nksauter@lbl.gov; asbrewster@lbl.gov; james.hohman@uconn.edu

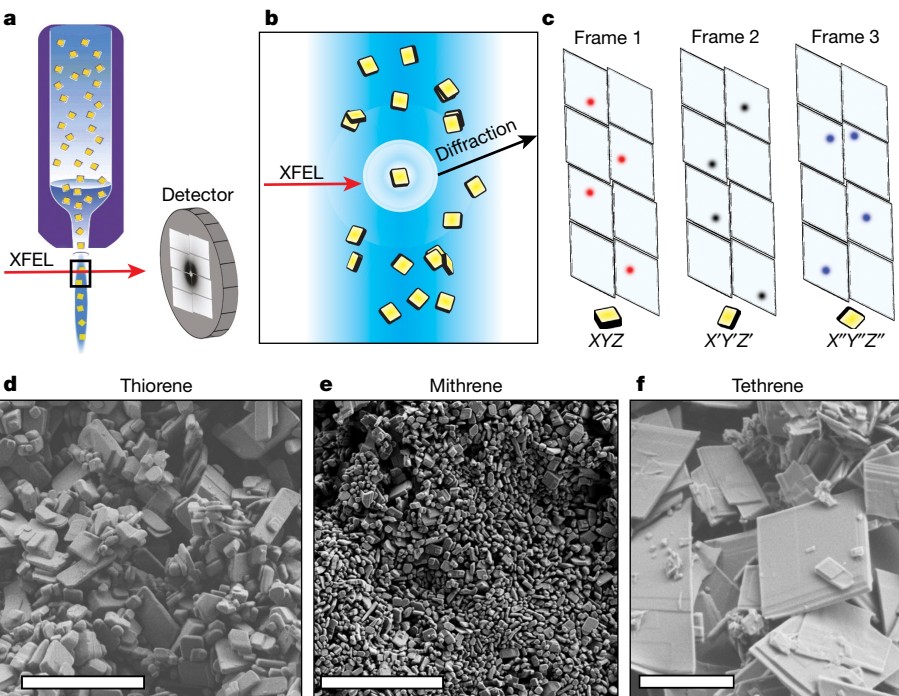

**Fig. 1 | smSFX experiment and the analytes of interest. a**, Schematic of the smSFX experiment. **b**, Randomly oriented microcrystals are delivered to the XFEL interaction point. 30-fs XFEL pulses interact with the microcrystals to produce diffraction images before destruction by the XFEL pulse. **c**, Individual frames from randomly oriented mithrene crystals are sparse. **d–f**, Scanning electron micrographs of all three silver benzenechalcogenolates, where some size heterogeneity and morphological divergence is noted. Scale bars are 5 µm.

solution step is often model-assisted, for example, by simulated annealing[24,25]. Three-dimensional electron diffraction (3D ED, also known as MicroED)[4–7] and serial electron diffraction[8–10] have also attracted attention for structure determination of nano- and microcrystals. Although this is generally performed under cryogenic conditions, some success has been reported using room-temperature 3D ED on radiation sensitive metal–organic frameworks with low-dose rotational scans[11]. SerialED has been successful in characterizing structures of zeolites[8,9] and proteins[10]. SerialED uses a single-shot-per-crystal approach, with sample preparation methods that mirror those of MicroED, where microcrystals are in vacuum conditions and affixed to a substrate. Here we present a complementary X-ray technique that also uses a single-shot-per-crystal approach. This method ensures intrinsic resistance to beam damage, requires almost no sample preparation, and is performed near room temperature and atmospheric pressure.

Serial femtosecond crystallography (SFX) usually involves a liquid jet of many (approximately $10^8$) small crystals injected into the interaction point of an X-ray free-electron laser (XFEL). The high-flux, femtosecond pulses available at XFELs enable a 'diffract before destruction' approach, yielding Bragg reflections in single-shot frames from individual submicrometre crystals. The femtosecond duration of data collection eliminates the role of beam damage[26]. The data collection occurs at room temperature using a liquid jet of suspended crystals, preserving the integrity of samples that may be sensitive to freezing or desolvation. Sample preparation involves loading a sample suspended in solvent into a pump syringe that is attached to a jet delivery system. This can be sample consumptive, but there are adaptations available to conserve sample and even recirculate the suspension through the interaction point[27]. High-speed data collection, currently in the kHz regime, is expected to give complete datasets in minutes of beam time[28].

There are several unique challenges for the application of SFX to chemical crystallography, generally centred on unit-cell determination and indexing diffraction patterns. A difference between traditional SCXRD and SFX is the unknown crystal orientation matrix. In a SCXRD experiment, the crystal is rotated on a goniometer, sequentially moving reflections into the diffracting condition. In SFX, each crystal is randomly oriented, so the orientation matrix must be determined independently for each frame (Fig. 1c). In macromolecular SFX, the large unit cell gives dozens of reflections per frame with obvious periodicity, permitting the use of Fourier methods for indexing[29,30] (Extended Data Fig. 1). In contrast to macromolecular SFX, we find that typical frames for a hybrid material have only 3–10 reflections (Extended Data Fig. 2). This sparsity requires the use of indexing algorithms that do not rely on periodicity of the input reflections[14,31]. We note that wider-band (pink-beam, Laue) methods can give more reflections per frame, a method applied to serial diffraction at synchrotron sources[31,32]. In this work, we develop methods that use the narrow bandwidth of the intrinsic XFEL spectrum, about 0.3% $\Delta E/E$. Previous work[14] has reported the partial structure of an amyloid peptide via SFX, applying the programme cctbx.small_cell to index small-molecule serial frames using a maximum clique algorithm that finds three-dimensional reciprocal space relationships in a sparse pattern. This algorithm requires a unit-cell candidate that must be discovered by other means.

In this work we present a technique for synthesizing a high-resolution powder diffraction pattern from XFEL data before generating unit-cell candidates by a custom adaptation of the SVD-Index algorithm developed for powder diffraction[33,34]. Using cctbx.small_cell indexing and room-temperature unit cells derived from the XFEL data, we index thousands of sparse partial diffraction patterns for mithrene, thiorene and tethrene. In all three cases we solve the crystal structures with no additional experimental information. The present work demonstrates that smSFX is a general technique for chemical crystallography on microcrystalline samples.

## smSFX experiment

An overview of the experiment is presented in Fig. 1. Crystals are suspended in a solvent and subjected to the XFEL beam. We arrive at a

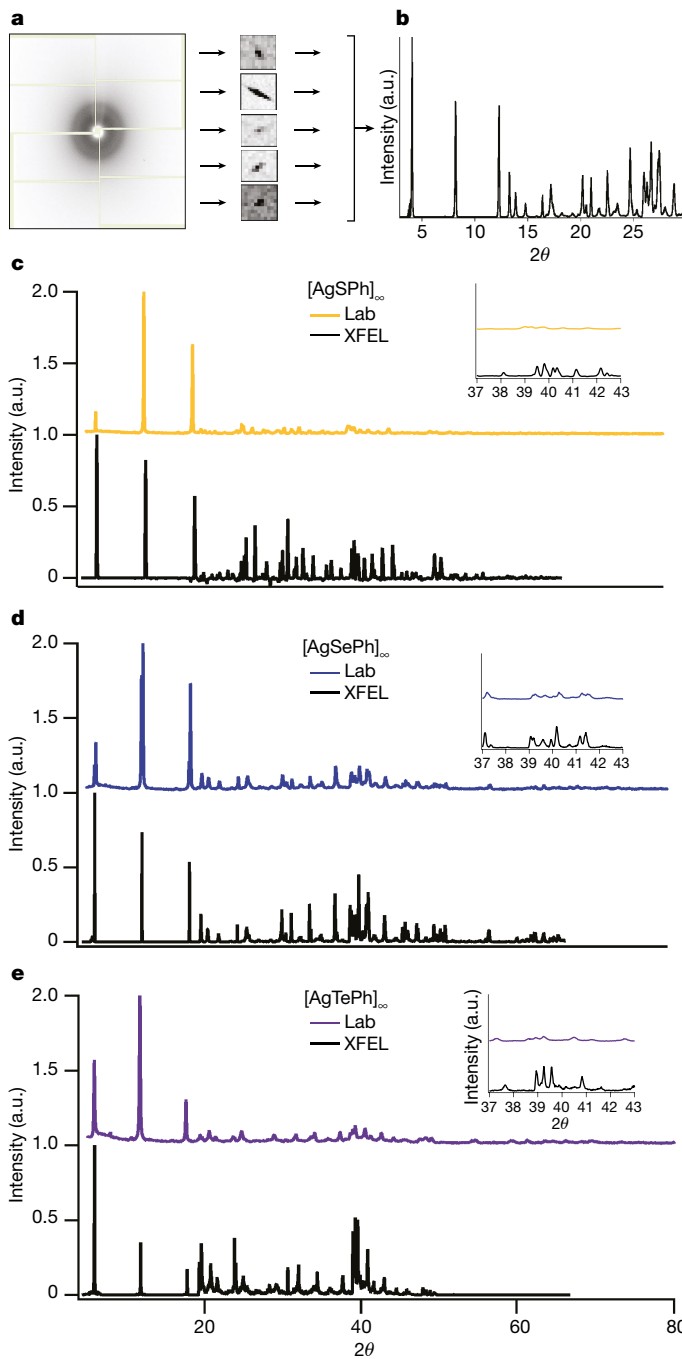

**Fig. 2 | Powder patterns derived from XFEL data. a**, **b**, A histogram over thousands of individual Bragg spot observations identifies the *d* spacings of the lattice, expressed here as the diffraction angle $2\theta$ (in °). As described in Methods section 'Synthetic powder patterns from XFEL shots', first the spots are found from the full dataset, then the spot positions are converted to *d* spacings. The *d* spacings are then binned into a histogram that amounts to a sharpened powder diffraction pattern. **c–e**, Radial averages of thiorene (**c**), mithrene (**d**) and tethrene (**e**) using the spot-finder radial histogram (black) compared to laboratory source XRD patterns (coloured). The horizontal axis is scaled to the wavelength of Cu Kα radiation ($\lambda = 1.5406$ Å), according to Bragg's law ($\lambda = 2d\sin\theta$), and the vertical axes are shown in arbitrary units. Thiorene was affected by an unknown impurity in low concentration and thus weak reflections have been downweighted. Insets show magnification of the data around $2\theta = 37°-43°$.

complete diffraction dataset by measuring approximately $10^6$ frames containing around $10^4$ indexable patterns. The morphology of the microcrystalline powders of the three compounds is shown in the electron micrographs in Fig. 1d–f. Each compound was produced in stoichiometric yield and in a single procedural step by heating a mixture of silver (I) oxide and benzenethiol or benzeneselenol, or by reaction of silver powder with diphenyl ditelluride in isopropanol. Purified crystals were then held either dry or suspended in solvent until the experimental beamtime.

We performed three experiments at XFEL facilities for proof of concept, method validation and new science. The proof-of-concept smSFX experiment at the Linac Coherent Light Source (LCLS)[12] involved an attempt to refine the mithrene crystal structure against the Cuthbert model[19] during a single 12-h shift. The resulting dataset permitted refinement with the known structure as a starting model, but an ab initio structure solution was not possible (further details are found in Methods). We then performed a follow-up experiment at SACLA[13] where 4.5 h of data collection (after optimizing sample delivery) enabled an ab initio structure determination of mithrene from an smSFX dataset. A third experiment at LCLS was performed to obtain smSFX datasets from thiorene and tethrene microcrystals, materials with no known structure. The higher repetition rate at LCLS enabled complete datasets to be collected for thiorene and tethrene in approximately 3 h each. This experiment proved the versatility of smSFX across XFEL sources and its ability to determine structures of true unknowns.

## Structure determination of mithrene

Unit-cell determination is the critical first step in solving a crystal structure, whether from SCXRD or SFX. In single-crystal diffraction, this task is accomplished using rotational scans and examining the periodicity of the lattice. In smSFX, the task is more difficult because no three-dimensional relationships between frames are immediately available. Following a previous approach[14] we looked to the field of powder diffraction, where several algorithms are available for indexing a list of one-dimensional *d* spacings. As these techniques generally require highly accurate *d* values, we developed a method with which to derive high-resolution powder patterns by synthesizing data over numerous XFEL diffraction patterns (Fig. 2). We used dials.find_spots[35] to locate spots with subpixel accuracy (Fig. 2a), eliminating sample-based and instrument-based peak broadening. Full details are given in Methods; the synthetic powder diffraction patterns are presented in Fig. 2c–e. Candidate unit cells were generated by a custom multiprocessed interface to the SVD-Index algorithm as implemented in the popular powder diffraction suite GSAS-II[33]. The candidate lattices were evaluated by agreement with the powder pattern and by indexing rate for frames from the XFEL dataset. The correct unit cell showed good agreement with the synthetic powder pattern and had the highest indexing rate using the cctbx.small_cell algorithm (Fig. 3e and Extended Data Fig. 3). We therefore correctly determined the room-temperature monoclinic unit cell of mithrene in agreement with the literature single-crystal structure[19].

Having determined the unit cell, we performed indexing, integration and merging using cctbx.small_cell_process and cctbx.xfel.merge. The space group was assigned by systematic absences and database frequency as is typical in chemical crystallography. Structure solution was performed by charge-flipping as implemented in olex2.solve[15]. This yielded a successful solution that revealed the heavy-atom substructure with Ag and Se distinguishable by their peak heights. At this stage, we discovered a simulated 'twin operator' that arose from a coincidental, pseudo-orthorhombic metric of the lattice. To resolve this, a second round of merging was performed using the heavy-atom partial structure as a reference for reindexing and scaling.

The model was completed by standard methods, using successive rounds of difference Fourier map calculation and atom assignment in ShelXL[16]. We were able to identify carbon atoms in difference maps (Fig. 3a, b). The $C_6H_5$ group was located in two positions disordered by rotation around the Se–C bond axis. Upon completion of the

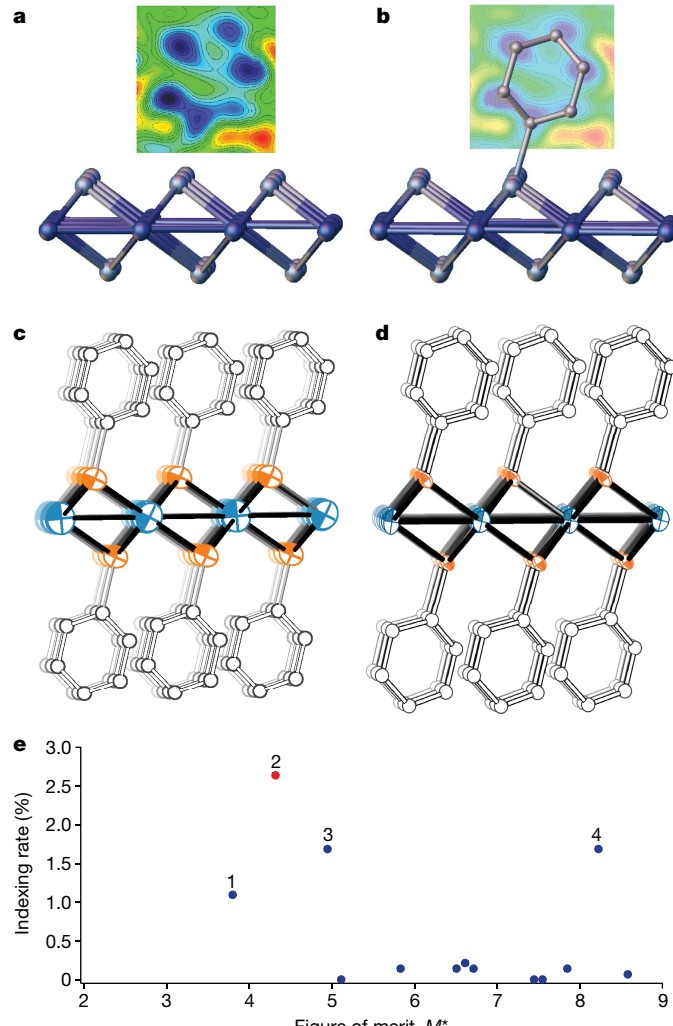

**Table 1 | Crystallographic results for thiorene, mithrene and tethrene**

| Compound | Thiorene | Mithrene | Tethrene |
|---|---|---|---|
| Formula | $AgSC_6H_5$ | $AgSeC_6H_5$ | $AgTeC_6H_5$ |
| Molecular weight | 217.03 | 263.93 | 312.57 |
| Space group | $Cc$ | $C2/c$ | $C2/c$ |
| $a$ (Å) | 7.290 | 5.938 | 5.900 |
| $b$ (Å) | 5.879 | 7.325 | 7.424 |
| $c$ (Å) | 28.072 | 29.202 | 30.258 |
| $\alpha$ (°) | 90 | 90 | 90 |
| $\beta$ (°) | 93.80 | 95.44 | 97.686 |
| $\gamma$ (°) | 90 | 90 | 90 |
| $V$ (Å³) | 1,200.6 | 1,264.3 | 1,313.3 |
| $Z$ | 8 | 8 | 8 |
| $\rho_{calc}$ (g cm⁻³) | 2.401 | 2.773 | 3.162 |
| XFEL source | LCLS | SACLA | LCLS |
| $\lambda$ (Å) | 1.24808 | 1.03207 | 1.24808 |
| $T$ (K) | 298 | 298 | 298 |
| $d_{min}$ (Å) | 1.35 | 1.2 | 1.35 |
| $\mu$ (mm⁻¹) | 16.19 | 10.68 | 32.56 |
| Frames | 1,426,595 | 492,509 | 1,187,208 |
| Crystals | 7,677 | 6,577 | 9,901 |
| Time (h) | 3.3 | 4.5 | 2.7 |
| Data | 265 | 372 | 289 |
| Restraints | 29 | 8 | 2 |
| Parameters | 53 | 35 | 29 |
| $R_1$ (obs) (%) | 10.5 | 11.3 | 10.6 |
| $R_1$ (all) (%) | 12.1 | 13.2 | 13.1 |
| $S$ | 1.10 | 1.12 | 1.13 |
| Peak, hole (e⁻ Å⁻³) | 0.94, −0.71 | 2.10, −1.67 | 1.93, −1.25 |
| $\langle\sigma(d_{Ag-E})\rangle$ (Å) | 0.039 | 0.010 | 0.011 |
| $\langle\Delta(d_{Ag-E})\rangle$ (Å) | n/a | 0.015 | n/a |

$\langle\sigma(d_{Ag-E})\rangle$ is the mean crystallographic uncertainty on silver-chalcogen distances; $\langle\Delta(d_{Ag-E})\rangle$ is the mean error as compared to the reference structure from SCXRD[19].

$Z$, formula units in unit cell; $\rho$, density; $\mu$, absorption coefficient; data, unique structure factors in refinement; $S$, goodness of fit.

n/a, not applicable.

**Fig. 3 | Mithrene structure determination results. a**, **b**, Identification of the $C_6H_5$ group in a Fourier difference map. The map is calculated from $F_{obs} - F_{calc}$ coefficients and contours are drawn at intervals of approximately 0.15 $e^-$ per Å³. The ring is disordered over two positions; both positions are shown in **b**, although they nearly overlap in this view. The plots were generated by Olex2[15]. **c**, **d**, Crystal structure of mithrene as determined in the present work (**c**) and the reference single-crystal structure[19] (**d**). **e**, Plot of candidate unit cells indexing rate versus $M^*$ (see equation (1) in Methods). Cell 2, labelled in red, is the correct cell, with the highest indexing rate and a low value of $M^*$.

model, we performed one final round of merging with this complete SFX-determined structure as a reference for scaling and reindexing. The final refinement converged with $R_1 (I > 2\sigma(I)) = 11.4\%$, where $I$-values are structure factor intensities and $\sigma(I)$ is the estimated uncertainty on $I$. Complete experimental and refinement statistics are given in Table 1.

Our approach of iterative structure refinement and re-scaling calls for some caution because there is a risk of biasing the data to match a model determined from a previous round of refinement. We demonstrated empirically (details in Methods) that the risk of model bias is not realized in these examples. In short, when a reference structure is 'sabotaged' by moving one atom a substantial distance, the final refinement returns that atom to its correct location; demonstrating that the intensities are not severely biased by the details of the reference model.

We validated the smSFX results by comparison to the literature structure determined by SCXRD (Fig. 3c, d and Table 1). The overall structures, including the disordered phenyl ring, are identical. The Ag–Se bonding distances were used as a quality metric. The mean

deviation from the reference was 0.015 Å, which is comparable to the least-squares uncertainty on the distances in the smSFX structure (Extended Data Table 1). Therefore, we conclude that the experimental errors are small and accurately estimated.

## Crystal structure of thiorene

Following the approach established for mithrene, we identified a base-centred monoclinic lattice for thiorene (Table 1). The lattice was similar to that of mithrene, but with the $a$ and $b$ axes exchanged. This suggested that the structure would be similar, with the symmetry elements differently placed. After indexing and merging, we solved the structure by charge-flipping in olex2.solve, initially in centrosymmetric $C2/c$. The basic structural motif, as predicted from the unit cell, is a distortion of the layered mithrene structure with $SC_6H_5$ groups each coordinating four Ag atoms. After extensive model-building and revision, with details given in Methods, we identified an ordered, pseudo-centrosymmetric model in polar $Cc$ with independent geometries for the top and bottom sides of each Ag–S sheet. The final refinement converged with $R_1 (I > 2\sigma(I)) = 10.5\%$ for data within the 1.35-Å limit imposed by the experimental geometry (Fig. 4).

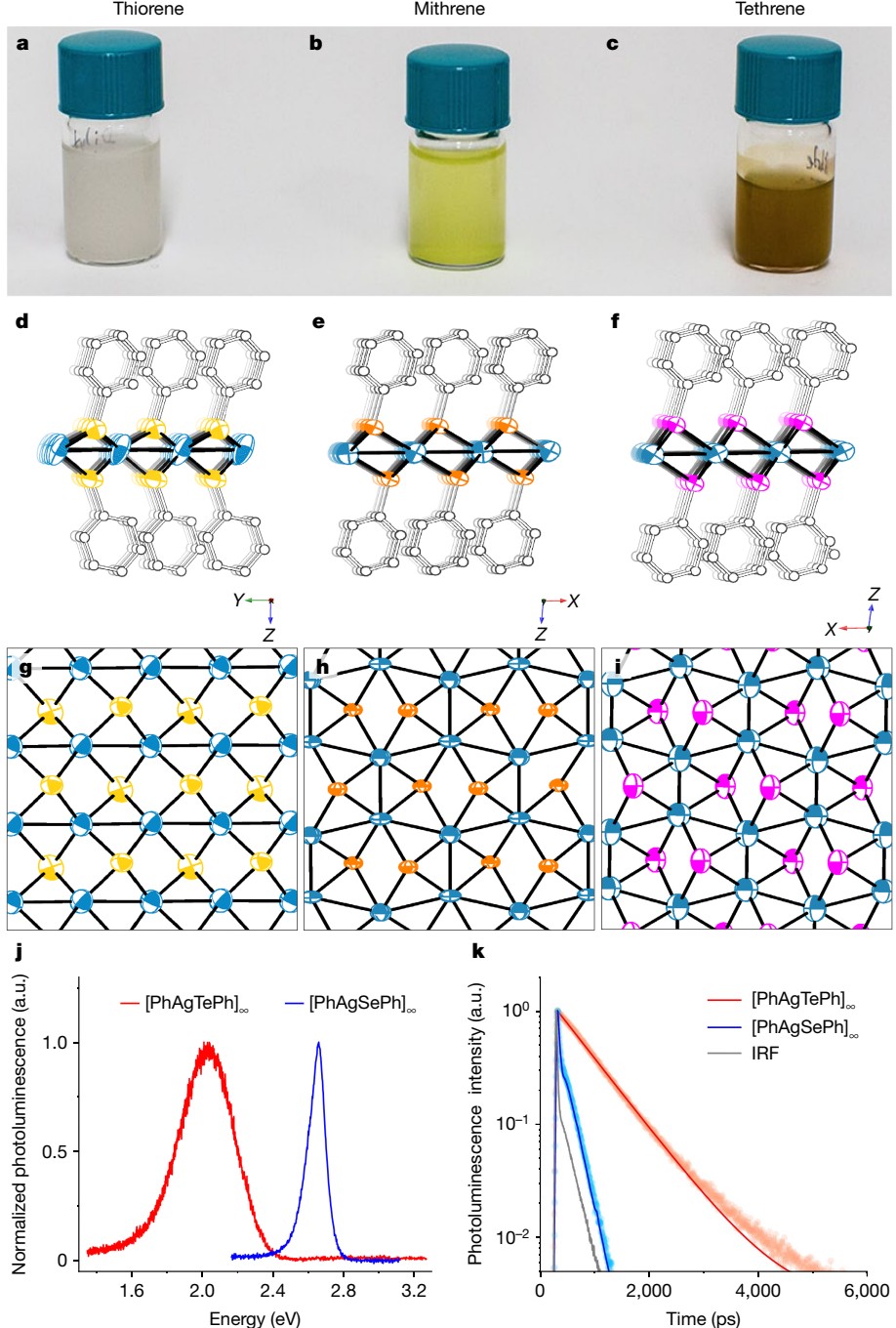

**Fig. 4 | Crystal structures, Ag–Ag motifs and optical properties of all three benzenechalcogenolates. a–c,** Suspended microcrystals of thiorene (**a**), mithrene (**b**) and tethrene (**c**) show their respective milky white, yellow and deep orange colours. **d–f,** Side and top views of crystal structures from smSFX for thiorene (**d**), mithrene (**e**) and tethrene (**f**). Thermal ellipsoids for Ag (blue), S (yellow), Se (orange) and Te (magenta) are drawn at the 50% probability level. Hydrogen atoms and one position of disordered $C_6H_5$ (for mithrene) are omitted for clarity. **g–i,** Models of thiorene (**g**), mithrene (**h**) and tethrene (**i**) with the view oriented down the *c* axis of the unit cell, with the carbon and hydrogen atoms omitted, displaying the divergence in the thiorene Ag–Ag bonding environment compared to that in mithrene and tethrene. **j,** Photoluminescence spectra of mithrene and tethrene under 467-nm excitation. Mithrene displays a sharp photoluminescence peak at approximately 2.7 eV; tethrene displays a broadband photoluminescence peak in the approximate range 1.5–2.4 eV. No photoluminescence is observed for thiorene. **k,** Time-dependent photoluminescence for mithrene and tethrene, with lifetimes of 50 ps and 1 ns respectively. IRF, instrument response function.

## Crystal structure of tethrene

We found a base-centred monoclinic lattice (Table 1), closely resembling the lattice of mithrene and thus probably isomorphous. After indexing and merging, we searched for solutions in *C*2/*c* using ShelXT[17] and obtained a solution for the heavy-atom substructure. As expected, the structure closely resembles mithrene, with small distortions to accommodate the longer Ag–Te bonds. The model was completed using a combination of difference maps and rigid fragments; full details are given in Methods. The final refinement had $R_1(I > 2\sigma(I)) = 10.6\%$ for data within 1.35-Å resolution (Fig. 4).

## Divergent optical behaviour explained

Mithrene and tethrene both have strong light–matter interactions; yellow mithrene has a direct bandgap and emits blue light with a short photoluminescent lifetime (approximately 50 ps);[18] whereas orange tethrene has an indirect bandgap[36] and emits broadly with a longer lifetime (approximately 1 ns) (Fig. 4j, k). By contrast, white thiorene has no emissive profile in the visible range. Our results give a concrete structural explanation for these observations. We have previously demonstrated that excitons in mithrene, delocalized in two dimensions across the argentophilic network of Ag–Ag bonds, give rise to its visible absorption and emission spectra[18,21,37]. The argentophilic interactions of thiorene are marked by linear Ag–Ag chains, which do not support the two-dimensional delocalization of excitons observed in mithrene and tethrene. Therefore, this smSFX study resolves the long-standing puzzle of the optoelectronic divergence of thiorene from its homologues mithrene and tethrene.

## Conclusion

XFEL smSFX can be used to solve structures of obligate microcrystals. We demonstrated this approach by using the cctbx.small_cell programme for indexing and integration of single-shot images from data collected on three microcrystalline materials. The resulting datasets were used to solve a 1.2-Å structure of mithrene, using a previous structure[19] only as a reference for method validation, and the 1.35-Å structures of thiorene and tethrene, with no reference structures. This approach should be broadly applicable across XFELs and synchrotron radiation facilities equipped for serial crystallography. Access to higher beam energies, with 25 keV expected at LCLS in 2022, will make resolutions of less than 0.7 Å available. Our results establish smSFX as a technique for microcrystal chemical crystallography that is free from beam damage. The intrinsically time-resolved nature of XFEL diffraction also presents opportunities for expansion into advanced structural studies.

*Note added in Proof*: After a recent report[38] of mithrene in $P2_1/c$, we checked our three datasets for possible primitive lattices. We confirmed that mithrene and tethrene are $C$-centred at room temperature. Cuthbert also reported a $C$-centred lattice for mithrene at 180 K (ref. [19]). Taken together, these observations imply a possible symmetry-lowering phase transition below 180 K in mithrene. For thiorene, there are weak violations of the $C$-centring condition, but we could not refine a primitive model using the systematically weak $h+k=2n+1$ reflections. Data collected at atomic resolution in the upcoming LCLS runs may yield further insights.

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

## Methods

### Synthesis

Benzeneselenol (97%, 375152), diphenyl ditelluride (98%, 384127), silver (I) oxide (99%, 221163), silver powder (>99.9%, 327077), and thiophenol (97%, T32808) were used as received from Sigma Aldrich. Microcrystalline silver benzeneselenolate (mithrene) and its homologues silver benzene thiolate (thiorene) and silver benzenetellurolate (tethrene) were prepared in an approach adapted from our earlier report[39]. Safety precaution: benzenethiol and benzeneselenol are toxic and foul-smelling, so adequate engineering protocols should be in place to prevent exposure. Benzeneselenol is susceptible to oxidation in air. Pure material should appear faintly yellow, whereas the presence of oxidized products (for example, diphenyl diselenide) will give a deeper orange colour, which generally will not impact the reaction until a substantial quantity of the benzeneselenol has oxidized. We endeavoured to plan our reactions such that the entire quantity of the benzeneselenol (1 g) was consumed, to eliminate the necessity of storing the opened containers.

For small-scale reactions producing mithrene and thiorene, 1 equivalent of silver (I) oxide (~50–150 mg) was placed in a 1-dram glass vial with 1.5 equivalents of benzeneselenol or benzenethiol. Reactions can be run solvent-free or in 1 ml of isopropanol. We find the addition of isopropanol is useful for subsequent sample recovery. The vial was sealed and heated in an oven to 70 °C. At elevated temperatures, reactions can appear complete within minutes; however, samples are allowed to incubate for 1 day to ensure stoichiometric yield. Caution when opening the vial is advised as the reaction is performed with excess benzeneselenol or benzenethiol, both of which are highly odorous and may be under pressure when hot. For our SACLA experiment, we used 3.0 g of silver oxide again with 1.5 equivalents of benzeneselenol in a 20 ml scintillation vial. The dramatic volume expansion of the silver oxide to the silver benzeneselenolate product caused the material to become impacted in the vial, so larger glassware may be advisable. Excess organic ligand is removed by 3 cycles of centrifugation (~10,000 rpm for 10 min), dilution with fresh isopropanol, and decanting. To prepare tethrene, 3:1 molar ratio of diphenyl ditelluride to solid silver (0) was added to a 4-dram vial with a PTFE-lined cap, along with 1 ml of isopropanol. The contents were thoroughly mixed and incubated at 80 °C in an oven for 4 days and recovered like the other examples. Product is transferred to a 45 ml Falcon tube and diluted with isopropanol. Products should not have a discernible odour. Crystals are stored refrigerated, when possible, protected from light, and either dry or suspended in methanol, ethanol or isopropanol.

Product purity was assessed by taking aliquots of the crystals from a slurry and depositing them on silicon substrates until dry. Laboratory powder X-ray diffraction experiments were sufficient to match powder patterns of earlier reports and to determine whether there was any silver oxide remaining. Images of all three benzenchalcogenolate crystals were imaged using a field-emission scanning electron microscope at an accelerating voltage of 1–5 kV.

### smSFX at SACLA

Having prepared of the order of 10[18] crystals, we used a Rayleigh jet injector system[40] at a high flow rate of ~150 μl min⁻¹ to reduce the possibility of clogging the capillary lines. The jet was positioned to intersect the 1 μm XFEL beam in atmospheric conditions at room temperature (25 °C). Diffraction patterns were recorded on the MPCCD detector[41] using 12 keV (1.032 Å) X-ray pulses with a duration of ~30 fs at a frequency of 30 Hz. We increased concentration in a stepwise fashion from 0.2 mg ml⁻¹ to 2 mg ml⁻¹ until we observed a hit rate of ~10%, which satisfied data-processing redundancy needs without causing capillary line clogs.

### smSFX at LCLS

Thiorene and tethrene were transported as dry solids to the Coherent X-ray Imaging endstation at the Linac Coherent Light Source[12] for smSFX experiments. Solvated samples were delivered to the XFEL interaction point via a 3D-printed polymer Gas Dynamic Virtual Nozzle (GDVN)[42] in high vacuum (10⁻⁸ torr). During prior experiments, we had used a methanol carrier solvent, but the polymer used in the 3D-printed GDVN was incompatible with alcohols. Urgent necessity required an alternate solution, and we quickly settled on a 0.2 wt% solution of Dawn dish detergent and water. This solution, affectionately labelled 'duck sauce' in honour of the friendly yellow duck featured on the product label, largely solved the issues with both clogging the capillary lines and compatibility with the GDVN. There was no evidence of reduced signal-to-noise values from inclusion of the detergent or contamination in the diffraction patterns. Sample concentration was adjusted to optimize hit rate as measured by spot-finding.

The XFEL beam was tuned to an average per-pulse photon energy of 9.8 keV with a repetition rate of 120 Hz. The per-pulse photon energy was measured using a downstream single-shot spectrometer[43]. Diffraction patterns were recorded on a Jungfrau 4M detector[44].

### Detector metrology

To synthesize virtual powder patterns from spot-finding results, highly accurate spot positions are required; therefore, the experimental geometry must be measured with high accuracy as well. We achieved this in a two-step process: first, the beam centre and detector distance were determined using powder diffraction rings from silver behenate; second, the internal detector metrology (relative positions of panels) was refined against diffraction from a high-quality protein reference sample.

Silver behenate powder samples were prepared by pressing Kapton tape into a dry powder. The tape was placed at the XFEL interaction point and exposed for ~5 min, after which the data were averaged together and dials.image_viewer was used to adjust the beam centre and detector distance.

For the metrology of the MPCCD detector at SACLA, protein diffraction data were collected in seven 10-min runs using crystals of Proteinase K from *Engyodontium album* (PDB reference 6k2x). For the detector at LCLS, we used four 5-min runs with crystals of hen egg-white lysozyme (PDB reference 3wun). The refinements were as described previously[45]. For the SACLA data we found it was necessary to refine individual panel rotations around all three axes. For the LCLS data the detector tilt was refined with all panels coplanar. The final spot position RMS deviations in the protein refinements were 14.1 μm (SACLA) and 26.6 μm (LCLS).

### Synthetic powder patterns from XFEL shots

We use the following approach to construct the powder pattern for unit-cell determination: (1) The DIALS spot-finder is used to harvest Bragg spots from the full dataset. This locates spot centroids with subpixel accuracy and thus eliminates peak broadening generated by sample (crystallite size, strain) and instrumental (axial divergence, detector point spread) factors. (2) The spot positions are converted to $d$ spacings using the per-shot measured XFEL photon energy. (3) The measured $d$ spacings are binned into a histogram that amounts to a sharpened powder diffraction pattern (Fig. 2b). This sharpening of the powder pattern is only possible because we observe a single crystal per shot, which differs from a conventional powder diffraction experiment. Peak resolution is often a limiting factor when attempting to identify a unit cell from powder data, so our technique gives a powerful advantage for cell determination. This technique is implemented as a new CCTBX script, cctbx.xfel.powder_from_spots.

### Unit-cell determination

Approximately 20 strong peaks were hand-selected from the powder pattern as input for indexing. Trials of GSASIIindex were performed for all Bravais lattices with a starting trial cell volume of 200 Å³. The obtained candidate lattices were ranked by a figure of merit:

$$M^* = \frac{\sum_{d_{min}}^{d_{max}} I \Delta d}{M_{20}},$$

where $M_{20}$ is as defined by de Wolff[46] and quantifies the normalized errors in $1/d^2$ of the peaks selected for the unit-cell search. The summation is over the points in the synthetic powder pattern, with $\Delta d$ calculated to the nearest peak predicted for the candidate cell. To complete the unit cell determination, we used the best 20 candidate unit cells (10 triclinic and 10 higher symmetry) to index diffraction frames by the cctbx.small_cell algorithm. The four unit cells that had the highest indexing rates, labelled in Fig. 3e, can be viewed in Extended Data Table 2. The correct unit cell indexed ~50% more diffraction frames than other candidate cells. Structure solution was attempted for the four candidates with reasonable indexing rates and was only successful for the correct cell. A utility for dispatching GSASII index trials and ranking the results is implemented in the new CCTBX script cctbx.xfel.candidate_cells.

## General methods for data processing

All processing of diffraction data was performed in a DIALS/CCTBX[35,47] environment, with code and instructions available from: https://github.com/cctbx/cctbx_project/tree/master/xfel/small_cell and links therein. After unit-cell determination as described above, indexing and integration were performed in cctbx.small_cell_process. An accurate reference geometry was provided for indexing; this was determined by refinement of recently collected diffraction frames from Proteinase K (SACLA data collection) or hen egg-white lysozyme (LCLS data collection). The space group for indexing was a symmorphic space group from the previously determined Bravais lattice, that is, space group $C2$ for the $C$-centred monoclinic lattices presented here. Crystal orientations were refined, but unit-cell refinement was disabled. It was necessary to decrease the fractional indexing cut-off from 0.3× the reciprocal axis length, optimized for protein diffraction, to 0.1×. This change has been included in cctbx.small_cell_process.

Merging was performed in a two-step process with no reference structure using cctbx.xfel.merge, a new version of the cctbx.xfel merging programme[29] that is designed for merging large numbers of images. First, the symmetry-equivalent measurements were merged using simple averaging. This dataset was then used as a scaling reference to re-merge the data, scaling each image to the reference set. This two-step approach was previously used to merge lysozyme XFEL diffraction data with accuracy that permitted de novo SAD (single-wavelength anomalous diffraction) phasing[48]. We note that in contrast to typical macromolecular SFX studies, the best results were obtained when we made no attempt to apply a separate resolution cut-off to individual frames on the basis of sparsity of spots on each image. Instead, the overall resolution cut-off was chosen to give 10–15× multiplicity for the outer shell in the final dataset. Errors in the merged reflection intensities were estimated using the standard error of the mean of the integrated reflections for each $hkl$, as described by protocol 1 published previously[48]. The results of this second merging trial were usable for structure solution by either olex2.solve[15] or ShelXT[17] as described in the individual refinement sections.

After the structure solution, we repeated the scaling and merging process using the working structure as a scaling reference. This is beneficial because the scaling model is known to be physical (atomic, with positive definite displacement parameters, and so on) in contrast to the previous reference intensities obtained by simple averaging. The scaling models were simplified with isotropic atomic displacement parameters (ADPs) fixed at reasonable values, typically 0.05 Å$^2$ for heavy atoms and 0.06 Å$^2$ for carbon. At this stage, we typically also tested pseudomerohedral reindexing operators as described in the individual refinement sections. We determined that X-ray absorption would be negligible ($T_{min} > 0.90$) for crystals in the 3–10 μm range; thus, no absorption correction was considered.

## Refinement of mithrene structure

The space group was assigned by standard methods. The presence of a $c$ glide plane was indicated by the systematic absences. Space group $C2/c$ was assigned tentatively because it is ~8× more common than $Cc$ for small-molecule structures. Charge-flipping in olex2.solve[15] yielded a successful solution, revealing the three independent atoms in the heavy-atom substructure, one Se atom on a general position and two Ag atoms on twofold special positions. Ag and Se were distinguishable by their peak heights. The heavy atom positions and isotropic ADPs were refined within a 1.2-Å resolution cut-off; this initial refinement converged with a poor $R_1$ agreement factor of 30.2%. The data were tested for twinning using PLATON[49] and revealed the twin operator (−1 0 0/ 0 −1 0/ 1 0 1), which was included in the refinement, improving the $R_1$ factor to 24.3%. We found that the apparent twinning was explained by a pseudo-orthorhombic metric (true: monoclinic C; $a, b, c$ = 5.94, 7.32, 29.20 Å; $\beta$ = 95.4°; pseudosymmetric: orthorhombic F; $a, b, c$ = 5.94, 7.32, 58.14 Å; $\alpha, \beta, \gamma$ = 90°, 90°, 89.56°). This metric pseudosymmetry presented an indexing ambiguity with some frames indexed incorrectly, thus simulating pseudo-merohedral twinning of the dataset. We applied the cctbx.xfel.merge tool modify_reindex_to_reference, reorienting individual frames by the twin operator to best match the reference structure.

After reindexing and merging with the heavy-atom substructure as a reference, our data were good enough to identify carbon atoms in difference maps (Fig. 3a, b). The $C_6H_5$ group was located in two positions disordered by rotation around the Se–C bond axis. In further refinement, the two disordered $C_6$ rings were each constrained to fit an ideal hexagon. Anisotropic ADPs were refined for Ag and Se; a single isotropic ADP was refined for the carbon atoms; hydrogen atoms were placed in calculated positions with riding isotropic ADPs. Finally, we performed one more round of merging with the full SFX-determined structure as a reference for scaling and reindexing. The final refinement converged with $R_1$ (all) = 13.3% and $R_1$ ($I > 2\sigma(I)$) = 11.4%.

## Refinement of thiorene structure

We initially solved the structure in $C2/c$ using olex2.solve and found that the heavy-atom positions formed a distorted analogue of the mithrene structure. The difference maps did not reveal the phenyl ring, so we tested several positions of a constrained $C_6$ ring with the aid of 1,2- and 1,3-distance restraints to regularize the geometry. No satisfactory placement was found; in all candidate orientations of the phenyl ring, there were nonbonded H–H clashes or severely distorted geometry around the S atom. Furthermore, there was no ordered model in $C2/c$ that gave agreement factors better than $R_1 \approx 19\%$ for all data within 1.35-Å resolution. We tested a disordered model for the $SC_6H_5$ group, which refined to a ~1:1 occupancy ratio with improved agreement factors, but it seemed unlikely that the entire bonding framework of the material would be disordered over two independent positions in a statistical ratio. Thus, we finally tested an ordered model in $Cc$ with the top and bottom sides of each AgS sheet independent from each other. This immediately improved $R_1$ to 14% for all data; furthermore, the geometry around S and the nonbonded H–H contacts within the thiorene sheets were all chemically reasonable. We placed hydrogen atoms in calculated positions and temporarily fixed all ADPs at reasonable isotropic values, 0.05 Å$^2$ for Ag, S and 0.06 Å$^2$ for C. This model was used as a final scaling reference; reindexing was also performed to attempt to resolve the orientation of the polar axis in $Cc$, although it is unlikely that this is resolvable for individual shots with current methods. The final refinement was performed with anisotropic ADPs (stabilized by restraints) for Ag and S and group isotropic ADPs for each of the phenyl rings. The phenyl rings were refined as rigid hexagons and the α-carbons were restrained to a chiral volume of 0. As with mithrene, we detected 'twinning' (misindexing) by a twofold rotation around the $c$* reciprocal axis, but with a smaller fraction of misindexed

frames (<5%). We note that the inversion pseudosymmetry gives larger uncertainties on the refined bonding distances than would otherwise be expected. The Flack parameter was not refined. The final refinement had $R_1$ (all) = 12.1% and $R_1$ ($I > 2\sigma(I)$) = 10.5%.

### Refinement of tethrene structure

After the initial structure solution and isotropic refinement, a difference map revealed the $\alpha$-C atom of the phenyl group, enabling us to locate the ring as a rigid fragment. After further refinement, the isotropic ADPs were fixed at typical values (0.05 Å$^2$ for Ag, Te, and 0.06 Å$^2$ for C). The resulting model was used as a reference for another round of scaling and merging using cctbx.xfel.merge to obtain a final dataset. Finally, hydrogen atoms were placed in calculated positions and anisotropic ADPs were refined for the Ag and Te atoms. The geometry of the phenyl ring was stabilized with restraints on C–Te 1,3-distances and on the chiral volume of the $\alpha$-carbon. The final refinement had $R_1$ (all) = 13.1% and $R_1$ ($I > 2\sigma(I)$) = 10.6%.

### Model bias test

Merging of mithrene diffraction data was performed using several different reference models; the structures were assessed by their root-mean-square deviation (RMSD) agreement (Ag and Se positions only) with the literature single-crystal structure[19] determined at 180 K (Extended Data Fig. 4, Extended Data Table 3). When reindexing and merging was performed with any reasonable model as a reference, the RMSD agreement of heavy-atom positions was excellent, around 0.015 to 0.02 Å. When the data were merged with no reference, either without or with a reindexing step, the RMSD agreement was slightly elevated to 0.025–0.03 Å. We explicitly tested for bias by 'sabotaging' a reference structure with a silver atom displaced by ~0.2 Å from its correct position. The resulting final structure had the silver atom displaced by a much smaller distance of 0.028 Å. Thus, although the reference model affects the final refinement, the refined parameters are determined mostly by the diffraction data and only slightly affected by the choice of reference model. The final SFX structure is in good agreement with the literature structure (Extended Data Table 1), with all bonding distances agreeing within 0.04 Å.

### Data availability

Crystallographic data for SFX-collected mithrene, thiorene and tethrene structures are available free of charge via the Cambridge Crystallographic Data Centre (CCDC), with deposition numbers 2059768, 2094416 and 2094415, respectively. Raw XFEL SFX data for mithrene, thiorene and tethrene are available free of charge via the Coherent X-ray Imaging Database (CXIDB), accession number 189, https://www.cxidb.org/id-189.html.

### Code availability

Code for generating powder patterns and interfacing with GSAS-II was implemented in a DIALS/CCTBX environment, with installation instructions available from https://github.com/cctbx/cctbx_project/tree/master/xfel/small_cell and links therein.

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

**Acknowledgements** We thank the SACLA and LCLS technical staff for their invaluable support. We are also grateful for the computational support from the SACLA HPC system. XFEL experiments were carried out at SACLA with the approval of the Japan Synchrotron Radiation Research Institute (JASRI) (proposal number 2019A8074). Research was supported by the NIH grant GM117126 to N.K.S. D.J.R. was supported by DOE-BER IDAT DE-AC02-05CH11231 and NIGMS ALS-ENABLE (P30 GM124169 and S10OD018483). D.M., J.P.B. and N.K.S. were supported by the Exascale Computing Project (grant 17-SC-20-SC), a collaborative effort of the Department of Energy (DOE) Office of Science and the National Nuclear Security Administration. E.A.S., A.S.B. and J.N.H. were supported by DOE ICDI grant DE-SC0022215. C.P.-R. and D.S.-I. were supported by PAPIIT IN216020 and CONACYT's CB-A1-S-8729. P.J.S. and K.Y. acknowledge support by Programmable Quantum Materials, an Energy Frontier Research Center funded by the US Department of Energy (DOE), Office of Science, Basic Energy Sciences (BES), under award DE-SC0019443. Use of the Linac Coherent Light Source (LCLS), SLAC National Accelerator Laboratory, is supported by the US DOE, Office of Science, Office of Basic Energy Sciences under contract no. DE-AC02-76SF00515. Work performed at the Molecular Foundry was supported by the Office of Science, Office of Basic Energy Sciences, of the US DOE under contract no. DE-AC02-05CH11231. This research used resources of the National Energy Research Scientific Computing Center (NERSC), a US DOE, Office of Science User Facility located at Lawrence Berkeley National Laboratory, operated under contract no. DE-AC02-05CH11231. B.H.Toby is acknowledged for assistance with GSAS-II.

**Author contributions** smSFX experiments were conceived and designed by E.A.S., D.W.P., A.S.B. and J.N.H. and were executed and analysed by E.A.S., D.W.P., R.B., D.M., J.P.B., A.B., D.J.R., R.G.S., M.H., A.A., B.H., C.P.-R., K.T., S.L., M.S., S.O., K.Y., P.J.S., T.K., D.S.-I., N.K.S., A.S.B. and J.N.H. Synthetic work was performed by E.A.S., R.P.K., D.C.P., M.Y. and J.N.H. The manuscript was written through contributions from all authors.

**Competing interests** The authors declare no competing interests.

**Additional information**
**Correspondence and requests for materials** should be addressed to Nicholas K. Sauter, Aaron S. Brewster or J. Nathan Hohman.

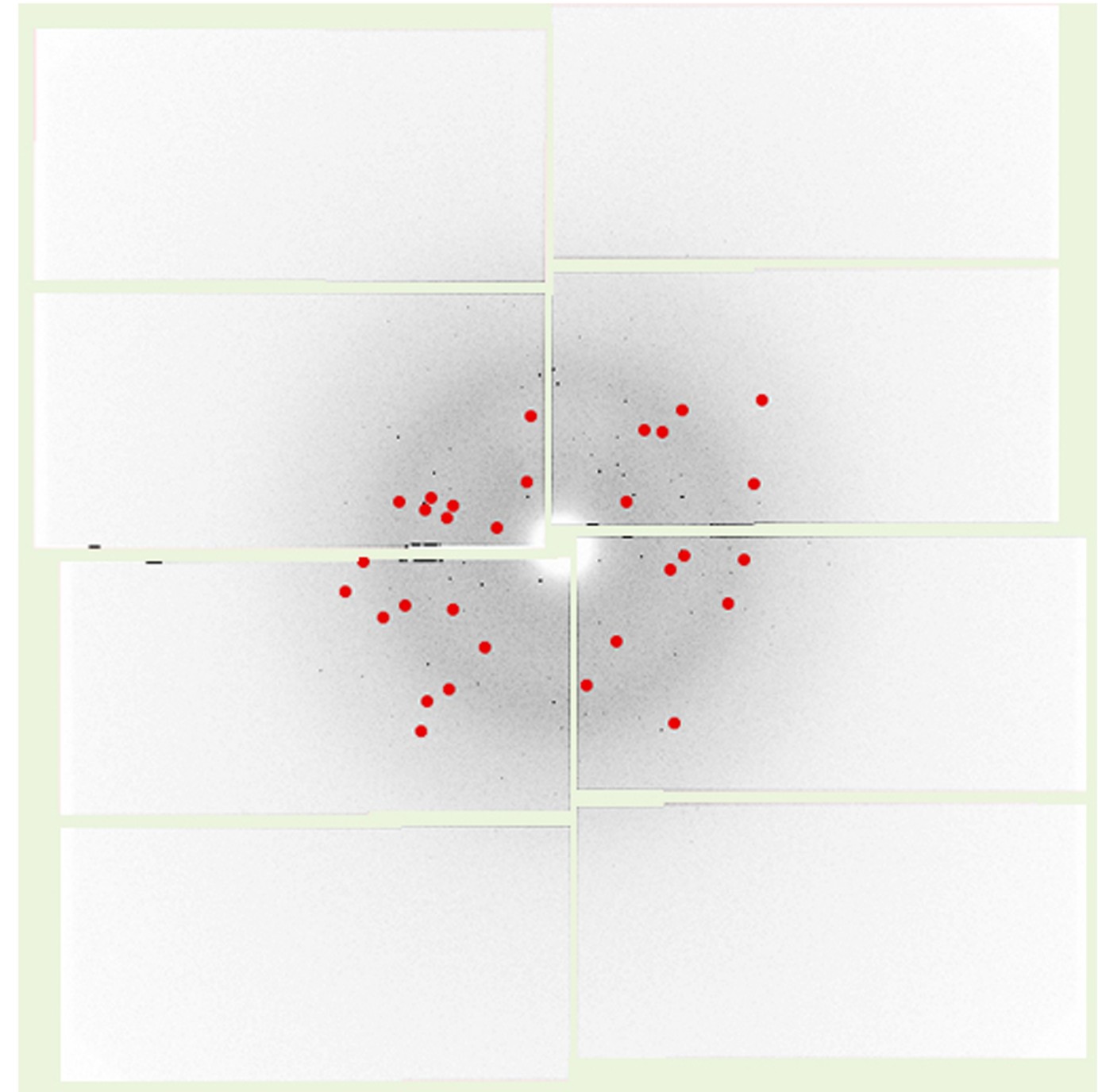

**Extended Data Fig. 1 | Single-shot frame from the Proteinase K dataset that was collected at SACLA for detector metrology on the MPCCD detector.** Indexed reflections are in red.

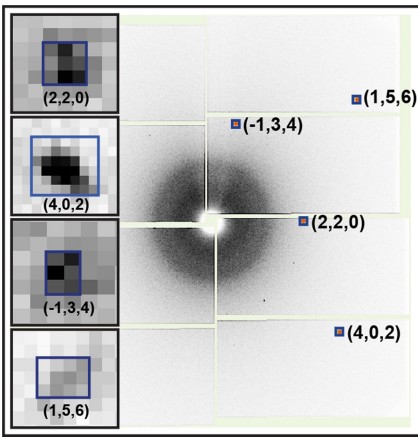

**Extended Data Fig. 2 | Single-shot frame from the mithrene dataset.**
Indexed reflections are boxed in the image and shown in the insets.

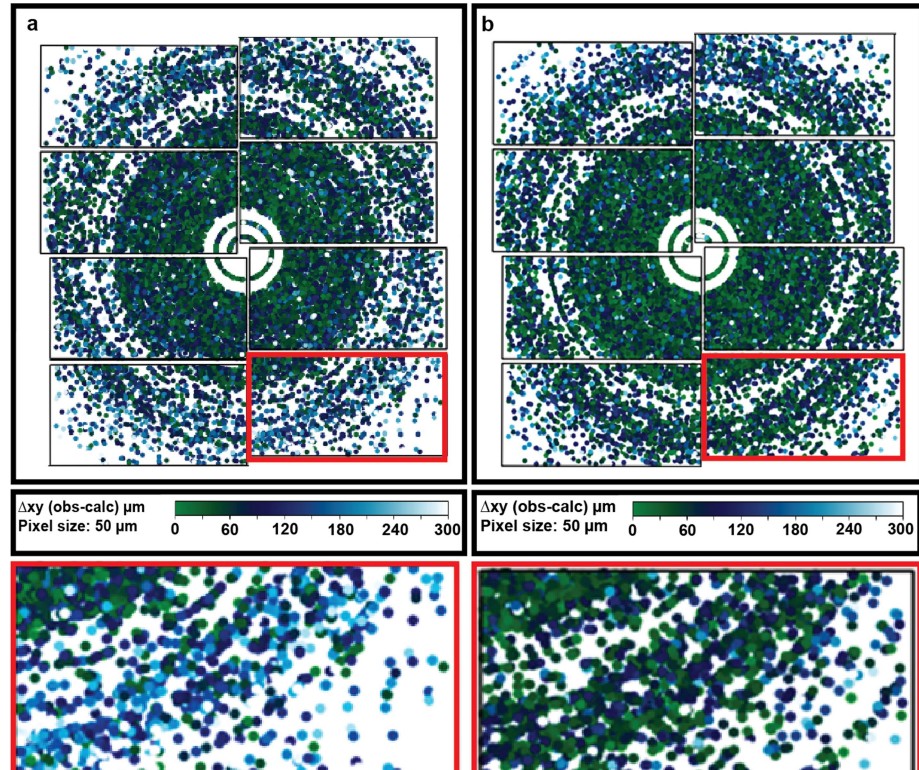

**Extended Data Fig. 3 | Indexing statistics using the XFEL-determined room-temperature mithrene unit cell. a**, Each spot from 6,577 mithrene crystals is shown on the detector simultaneously, coloured by the magnitude of the difference (in μm) between predicted and observed spot locations on the detector. Using the literature value cryogenic unit cell[19] to index the whole XFEL dataset produced an overall RMSD between predicted and observed reflections of 123.3 μm. **b**, Using the XFEL determined the room-temperature unit cell improved indexing accuracy, with an overall RMSD of 110.6 μm and increased the number of indexed high-resolution reflections, seen in the insets of the bottom corner panel on the detector.

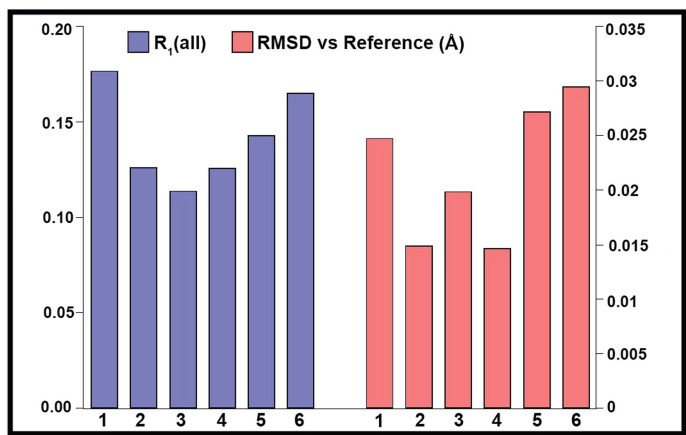

**Extended Data Fig. 4 | Iterative reindexing and merging with reference model bias test results.** $R_1$ is calculated for all data. RMSD is the deviation of heavy atom (Ag, Se) positions in angstroms, from the reference structure in ref.[19].

**Extended Data Table 1 | Interatomic distances in mithrene measured by SFX and SCXRD[19]**

| Atom 1 | Atom 2 | Bond Lengths (Å) (SFX) | Bond Lengths (Å) (SCXRD) |
|---|---|---|---|
| Ag1 | Se3 | 2.823(11) | 2.7816(14) |
| Ag1 | Se3a | 2.689(9) | 2.6939(12) |
| Ag1 | Ag2 | 2.907(10) | 2.9110(16) |
| Ag1 | Ag2b | 3.063(3) | 3.029(5) |
| Ag2 | Se3a | 2.725(9) | 2.6980(12) |
| Ag2 | Se3c | 2.770(11) | 2.7757(14) |
| Se3 | C1 | 1.95(3) | 1.911(8) |

**Extended Data Table 2 | Candidate mithrene unit-cell dimensions, where cell 2 is the correct room-temperature unit cell**

| Cell | $a(Å)$ | $b(Å)$ | $c(Å)$ | α(°) | β(°) | γ(°) | Bravais Lattice |
|------|--------|--------|--------|------|------|------|-----------------|
| 1 | 5.93562 | 7.32614 | 14.5982 | 90 | 95.439 | 90 | P2 |
| 2 | 5.93762 | 7.32461 | 29.202 | 90 | 95.4404 | 90 | C2 |
| 3 | 5.93672 | 7.32923 | 14.5979 | 90 | 95.4236 | 90 | C2 |
| 4 | 4.76907 | 22.7631 | 14.6143 | 90 | 96.1409 | 90 | C2 |

**Extended Data Table 3 | Descriptions of merging protocols for mithrene structure determination and model bias test**

| Trial | Protocol |
|---|---|
| 1 | No reindexing, no reference model |
| 2 | Reindex and scale vs. Ag, Se substructure from charge-flipping |
| 3 | Reindex and scale vs. full SFX structure. |
| 4 | Reindex and scale vs. single crystal XRD structure |
| 5 | Reindex and scale vs. "sabotaged" structure. (Purposeful displacement of a silver atom by ~0.2 Å) |
| 6 | Multiple rounds of reindexing and scaling without a reference structure |