## [Peer Review File · Nature]

Manuscript Title: Chemical Crystallography by Serial Femtosecond X-ray Diffraction

Reviewer Comments & Author Rebuttals

Reviewer Reports on the Initial Version:

Referee #1 (Remarks to the Author):

The manuscript describes the first structure determination of a "small molecule" using serial femtosecond X-ray crystallography. The resolution obtained for the structure of silver benzeneselenolate (mithrene) was 1.2 Angstroms and the structure is essentially the same as that elucidated by conventional single-crystal X-ray crystallographic techniques, which confirms the effectiveness of the technique. The key issue that is addressed is that in small molecule studies, because the crystallographic unit cells are so much smaller than in protein crystallography (where serial femtosecond crystallography has been used successfully for a number of years) there are only a small number of reflections in a given diffraction image and, as a result, the indexation to obtain the unit cell is difficult, and without knowledge of the unit cell the structure cannot be determined. Using the SFX method with a large number of crystals and state-of-the-art software to deconvolute the images that contain many diffraction spots from different crystals (akin to a powder diffraction image) the authors were able to obtain the unit cell, index the diffraction pattern, and solve and refine the crystal structure using conventional small molecule single crystal techniques.

The work represents a major advance in the SFX area and one that has an enormous potential if applied to the plethora of small molecule structures. Currently, tens of thousands of new small molecule structures are determined each year. This technique could revolutionise the approach to small molecule crystallography and eliminate much of the need to grow larger (10 micron) sized crystals. The reviewer is unaware of any other results of this type and the results reported here represent a significant breakthrough.

That said there is still a long way to go. Small molecule crystallographers normally obtain data of sub-Angstrom resolution in order to obtain accurate structures with low estimated standard deviations. There are also issues with absorption effects from the heavier atoms (Silver and selenium in this case) that will need to be addressed, although the use of multiple crystals will partly overcome this.

The methodology used is exciting and certainly state of the art. It builds on Laue diffraction work carried out at synchrotrons and also a neutron sources. So there is a good chance that it will be generally applicable.

The conclusions are valid and the references appropriate.

My only factual question on the data collection and analysis concerns the SACLA data that required 36 hours to collect. This seems a long time and this statement requires further clarification in the main text.

There is a case for publication of this work in Nature as a "first" but there is also a case that the current work still has a long way to go and it might be a little niche for a journal with a remit as broad as Nature, and perhaps Nature Methods or Nature Communications would be more suitable at this point?

Referee #2 (Remarks to the Author):

The manuscript "De Novo Small-Molecule Structure Determination of a Hybrid Material by Serial Femtosecond Crystallography" by E. A. Schriber et al. describes data collection and structure solution of a mithrene from a free electron laser experiment. According to the authors, this is the first structure solved

ab initio from FEL data. To the best of my knowledge, this claim is correct, which makes this work highly interesting.

It may be the basis for time-resolved reaction studies from solid state chemicals (in crystalline form). As such, this manuscript may have the same impact as the first SAD experiment from FEL data (Barends et al, Nature 2014). Therefore, Nature is a suitable journal for the publication of this work.

The manuscript is well written. The experimental steps, as well as the validation steps are clear and comprehensible. The authors took measures to prevent over-interpretation or misinterpretation of their results and describe caveats and alternatives. The most important question is whether these data contain structural information. This question is confirmed with Fig.4, which shows the light atom benzene ring in the difference map after three heavy atoms (Se/Ag) could be placed with solvent flipping.

The manuscript does not hide the fact that data interpretation has been difficult. Small molecules only exhibit very few reflections per FEL diffraction pattern, and indexing is a major obstacle. This part is described in good detail with comprehensible details. As one would expect, quality indicators are higher than for the typical single crystal X-ray structure, but all within reasonable ranges. Some statistics are better than I would have expected, like CC1/2, Rint, R1.

There are several technical aspects that would improve the manuscript. Some of these aspects may not be realisable, as FEL experiments are rare and data volume can be enormous.

- The indexing produced four candidate cells. The authors picked the one with the highest indexing rate, but this rate is only 2.5% compared to 1-1.5% for three other candidates. Could the authors integrate the data with all four cells, make structure solution attempts and, where successful, compare the quality of the resulting structure? With this, the authors could indeed claim that they determined the unit cell rather unbiased.

- The authors did not make use of an X-ray powder pattern. Why? There is sufficient material, and one could have used this information. Instead, the experimental FEL data were turned into a powder pattern and treated as such. Could the whole indexing procedure be simplified by combining with the information from an X-ray powder pattern? The amount of material necessary for an FEL experiment is certainly sufficient to collect an X-ray powder pattern with a dedicated powder instrument.

- The authors mention the energy spread of the FEL. Can they quantify the spread? Can it be shown as a histogram over energy?

- The authors first used a 'maximum value' search for the Bragg peaks, but this produced no useable result. Why did they not use the dials.spot-finder from the beginning? It seems that the spots were normalised by their energy, i.e. converted to d-spacing. This seems an obvious way to go. Is this not so obvious?

- The authors cite N. K. Sauter (2015) for improving data intensities, but did not make use of this improvement. Why not?

- Data were integrated to 1.15Å, but for refinement, data were cut to 1.2Å. Why? This throws away more than 10% of the data.

- Similar to (1): The authors indicate a space group ambiguity between C2/c and Cc. In such cases, one would continue with both until a discrimination is clear, either because the structure can only be solved in one space group, or because refinement yields better results. This step should be included. Sometimes, of course, one cannot decide between space groups. Also in this case, data should be presented.

- What is the standard uncertainty in the cell parameters?

Minor technicalities:

- The title should be changed from "De novo" to "Ab initio". "Ab initio" (cf. Int. Tables Cryst. Vol F) in crystallography refers to phasing based only on intensities, which is the important point of this work.

- The term 'mithrene' appears the first time on p. 3, line 75, but is used there like it has been introduced before (unless 'Summary' is considered part of the main text).

- p. 3, line 79 "... shown in the supporting materials". A reference to Fig. S1, S2 should be added here.

- p. 4, around line 89: "results in few (3-10) reflections per frame": this is a repetition of p. 3, line 75.

- The abstract describes mithrene to have 9 independent non-H atoms. In the context of the manuscript, this is exaggerated, as only 4 independent non-H atoms were refined, plus the shift and rotations of the benzene ring. However, with mild restraints (SADI for 1,2- and 1,3-distances of the C6-ring), all atoms can indeed be refined independently, which the authors may want to do.

- Fig. 4C: the comparison with the SXR structure (Cuthbert et al) does not show very interesting details. One could instead use this panel to show the disordered benzene ring down the C-Se bond. This would be more informative.

Referee #3 (Remarks to the Author):

The manuscript presents a determination of the crystal structure of mithrene by means of serial femtosecond crystallography (SFX). The key claim of the manuscript is that it demonstrates the feasibility of determination of small-unit-cell materials by SFX.

The aim of this review is two-fold: to assess the technical quality of the structure determination and correctness of the claims in the manuscript, and to evaluate the novelty, originality and importance of the work.

Concerning the first point, I can say that, technically speaking, the goal of the manuscript was fulfilled and the demonstration that small crystal structures can be solved by SFX was successful. My more detailed comments on certain questionable aspects of the structure analysis follow at the end of the report.

It is more difficult to assess the aspects of novelty and importance. A manuscript can be groundbreaking either thanks to the novelty of the presented approach, by the importance of the results or by the (potential) impact of the presented procedures or methods. Unfortunately, as much as I regret it, I must say that I fail to see the groundbreaking nature of the presented manuscript in any of the three ways. Let me explain this opinion more specifically:

(1) novelty of the presented approach: the manuscript seems to be combining experimental and computational procedures known and presented before. The difference between "large molecule" and "small molecule" SFX is not fundamental, and "small-molecule" SFX just needs to overcome several specific computational difficulties, mostly related to the sparsity of individual diffraction patterns. Most of these difficulties, if not all, have been already solved elsewhere. The manuscript itself notes the work of Brewster et al. that used the same algorithms for the key indexing step. The manuscript, however, does not reference e.g. the work of Dejoie et al. (Serial snapshot crystallography for materials science with SwissFEL, IUCrJ 2, 361-370, 2015). Although this work mimicks the SFX data using standard synchrotron beam-line and the results are thus somewhat artificial, it addresses most of the problems (sparseness of the patterns, indexing, merging...). Another work addressing indexing of sparse small-molecule frames is Dejoie nad tamura, J. Appl. Cryst. (2020). 53, 824-836. Moreover, a lot has been done on serial electron crystallography of small (non-macromolecular) materials (e.g. <http://scripts.iucr.org/cgi-bin/paper?S1600576717005854>), and this work is also neither mentioned nor referenced in the present manuscript (see also below for more comments on electron crystallography).

(2) importance of the specific results: the structure of mitherene is known and it is not the ambition of the manuscript to present a new important structure. This aspect is thus also not relevant here.

(3) potential impact of the presented method: SFX is only one of several methods to solve structures from microcrystalline samples. Two other prominent examples are x-ray powder diffraction and electron single crystal diffraction. In the manuscript very little is dedicated to the comparison of the presented method to the "competition". It seems to me that the amount of material needed is relatively large and thus it should be possible to collect a standard powder x-ray diffraction (PXR) pattern from it. At this size of the unit cell a solution from powder diffraction would most likely not pose big problems. Where PXR fails is either for phase mixtures - but SFX suffers from the problem of phase mixtures as well - or larger unit cells, but there the SFX approach moves closer to the well established "large unit cell" SFX methods. Even more important is the comparison to 3D electron crystallography methods (3D ED, aka MicroED). The manuscript deals with electron diffraction in a single sentence, stating that "electron diffraction is not easily applied when samples are radiation sensitive or experimental conditions are near standard temperature and pressure". This is true only in the last aspect - electron diffraction cannot be performed on crystals at ambient pressure. Otherwise, numerous studies have been done on very radiation sensitive materials both at ambient and low temperature. Moreover, serial electron diffraction method (SerialED) has been developed and is in use, which overcomes most of the remaining difficulties with radiation sensitive materials. Thus, electron diffraction is a relatively (in comparison to SFX) easily accessible method solving most of problems small-molecule SFX would be attractive for. I thus see very little room for problems in small molecule crystallography that cannot not be tackled by other, more accessible and less involved methods, and would require the use of SFX techniques.

In summary, proving that SFX can be successfully used to solve a small-molecule structure is certainly interesting, it is opening new possibilities and enriching the landscape of available methods to deal with microcrystalline materials, but I fail to identify the necessary aspects that would make this development a significant breakthrough.

Getting back to the actual contents of the manuscript, here are a few more specific comments:

- line 43: the references to electron diffraction works are not very representative. If structure solution of small-molecule compounds by electron diffraction is referenced, then either the primary works by Kolb et al. should be referenced (possibly together with the more recent works by Jones et al. - reference 3, and Gruene et al. -

<https://onlinelibrary.wiley.com/doi/full/10.1002/anie.201811318>), or the big review by Gemmi et al., which summarizes most of the relevant literature before 2019:

<https://pubs.acs.org/doi/abs/10.1021/acscentsci.9b00394>.

Twinning: I am not convinced by the description and argumentation supporting the discussion about twinning. If there was a full indexing ambiguity, then the twin ratio would be expected to be

close to 50:50. If the ambiguity would be present only for some orientations and not for other, then using an overall twin law would not be the appropriate way to describe the situation. In any case, when the preliminary model was used to re-index the data and remove the indexing ambiguity by correlation with the calculated intensities, then the twin ratio should gradually shift towards 100:0. However, in reality it remains constant and only in the last step changes slightly from 81:19 to 85:15. This does not make sense to me and makes me believe that there is another effect involved. Have you considered the possibility that individual microcrystals can be twinned and the data themselves come from twinned crystals? If this is the case, then the whole analysis would be complicated, because the twin ratio for each crystallite may be different. See also my next comment.

Disorder in the structure: You found disorder in the phenyl ring, while the reference structure (Cuthbert et al.) does not report this disorder. This discrepancy is not even mentioned in the manuscript, and there is no discussion about the disorder. The disorder is not enforced by symmetry and it is thus also not obvious why the ratio of the two orientations should be 50:50. Have you tried to refine the ratio? The structure is layered and I thus suspect it could be prone to polytypism. The (possible) twinning points also to this direction. Could it be that the disorder is in reality not disorder, but rather an unaccounted-for twinning? Maybe the observed disorder and twinning are just two demonstrations of the same effects. I think that a more detailed analysis of these aspects is necessary.

Author Rebuttals to Initial Comments:

Referee #1 (Remarks to the Author):

The manuscript describes the first structure determination of a "small molecule" using serial femtosecond X-ray crystallography. The resolution obtained for the structure of silver benzeneselenolate (mithrene) was 1.2 Angstroms and the structure is essentially the same as that elucidated by conventional single-crystal X-ray crystallographic techniques, which confirms the effectiveness of the technique. The key issue that is addressed is that in small molecule studies, because the crystallographic unit cells are so much smaller than in protein crystallography (where serial femtosecond crystallography has been used successfully for a number of years) there are only a small number of reflections in a given diffraction image and, as a result, the indexation to obtain the unit cell is difficult, and without knowledge of the unit cell the structure cannot be determined. Using the SFX method with a large number of crystals and state-of-the-art software to deconvolute the images that contain many diffraction spots from different crystals (akin to a powder diffraction image) the authors were able to obtain the unit cell, index the diffraction pattern, and solve and refine the crystal structure using conventional small molecule single crystal techniques.

The work represents a major advance in the SFX area and one that has an enormous potential if applied to the plethora of small molecule structures. Currently, tens of thousands of new small molecule structures are determined each year. This technique could revolutionise the approach to small molecule crystallography and eliminate much of the need to grow larger (10 micron) sized crystals. The reviewer is unaware of any other results of this type and the results reported here represent a significant breakthrough.

We appreciate this comment. The tethrene and thiorene structures presented in this revision are good examples of the problems that will be solved by smSFX.

That said there is still a long way to go. Small molecule crystallographers normally obtain data of sub-Angstrom resolution in order to obtain accurate structures with low estimated standard deviations. There are also issues with absorption effects from the heavier atoms (Silver and selenium in this case) that will need to be addressed, although the use of multiple crystals will partly overcome this.

Regarding the resolution, the LCLS-II-HE upgrade, available to users around the beginning of 2022 during the next beamtime cycle, is expected to provide hard X-rays in the 20 keV/0.62 Å range. With the experimental geometry presented in this work, we would obtain 0.68 Å resolution with 20 keV photons.

Regarding absorption, we determined that in the present work the transmission factors are >0.9 for crystals under 3 μm (for tethrene at 9.5 keV) to 10 μm (for mithrene at 12 keV). Since SEM micrographs revealed that the three microcrystalline samples here were mostly in the <1 μm range, we consider it reasonable to neglect the effects of absorption. We note that the simplest sign of uncorrected absorption, pathologically small displacement factors, was not observed in any of these structures.

The methodology used is exciting and certainly state of the art. It builds on Laue diffraction work carried out at synchrotrons and also a neutron sources. So there is a good chance that it will be generally applicable.

The conclusions are valid and the references appropriate.

My only factual question on the data collection and analysis concerns the SACLA data that required 36 hours to collect. This seems a long time and this statement requires further clarification in the main text.

We added a table with data collection and refinement statistics. The actual measurement time for mithrene at SACLA was updated to 4.5 h (500k frames at 30 Hz). The balance of the 36 hours was spent on optimizing sample delivery; we feel that the actual data collection time is a more important figure for considering the future viability of this technique. The beam time required was already improved to ~3 h using 120 Hz data collection at LCLS. MHz data collection is becoming available at EuXFEL, see: <https://iopscience.iop.org/article/10.1088/1748-0221/10/01/C01023>.

There is a case for publication of this work in Nature as a "first" but there is also a case that the current work still has a long way to go and it might be a little niche for a journal with a remit as

borad as Nature, and perhaps Nature Methods or Nature Communications would be more suitable at this point?

We feel that the inclusion of two previously unknown structures, including the revision of the thiorene structure proposed by Dance et al [1991], shows that smSFX is already a viable technique that is making important contributions.

Referee #2 (Remarks to the Author):

The manuscript "De Novo Small-Molecule Structure Determination of a Hybrid Material by Serial Femtosecond Crystallography" by E. A. Schriber et al. describes data collection and structure solution of a mithrene from a free electron laser experiment. According to the authors, this is the first structure solved ab initio from FEL data. To the best of my knowledge, this claim is correct, which makes this work highly interesting.

It may be the basis for time-resolved reaction studies from solid state chemicals (in crystalline form). As such, this manuscript may have the same impact as the first SAD experiment from FEL data (Barends et al, Nature 2014). Therefore, Nature is a suitable journal for the publication of this work.

The manuscript is well written. The experimental steps, as well as the validation steps are clear and comprehensible. The authors took measures to prevent over-interpretation or misinterpretation of their results and describe caveats and alternatives. The most important question is whether these data contain structural information. This question is confirmed with Fig.4, which shows the light atom benzene ring in the difference map after three heavy atoms (Se/Ag) could be placed with solvent flipping.

The manuscript does not hide the fact that data interpretation has been

difficult. Small molecules only exhibit very few reflections per FEL diffraction pattern, and indexing is a major obstacle. This part is described in good detail with comprehensible details. As one would expect, quality indicators are higher than for the typical single crystal X-ray structure, but all within reasonable ranges. Some statistics are better than I would have expected, like CC1/2, Rint, R1.

There are several technical aspects that would improve the manuscript. Some of these aspects may not be realisable, as FEL experiments are rare and data volume can be enormous.

- The indexing produced four candidate cells. The authors picked the one with the highest indexing rate, but this rate is only 2.5% compared to 1-1.5% for three other candidates. Could the authors integrate the data with all four cells, make structure solution attempts and, where successful, compare the quality of the resulting structure? With this, the authors could indeed claim that they determined the unit cell rather unbiased.

We appreciate this comment and we have proceeded with indexing, integration and merging for the top four candidates. Candidate 2 (the correct one) is solved by charge-flipping in Olex2 with a ~50% success rate; the proposed structure for this 50% of "hits" is always the same. The other three candidates occasionally give a proposed solution, but the results are always obviously unreasonable and can be easily rejected.

- The authors did not make use of an X-ray powder pattern. Why? There is sufficient material, and one could have used this information. Instead, the experimental FEL data were turned into a powder pattern and treated as such. Could the whole indexing procedure be simplified by combining with the information from an X-ray powder pattern? The amount of material necessary for

an FEL experiment is certainly sufficient to collect an X-ray powder pattern with a dedicated powder instrument.

The "powder pattern" synthesized from XFEL frames is special because we obtain a sharpening effect by observing single reflections (resolved in three dimensions) one by one. We locate peak centroids with sub-pixel accuracy and thus avoid the broadening generated by sample-based (crystallite size, strain) and instrumental (axial divergence, detector point spread) peak broadening. We also avoid adding inter-peak diffuse background to the signal by only including peak centroids, instead of summing many patterns together, including background. This sharpening is only possible because we observe a single crystal at a time, which differs from a conventional powder diffraction experiment. Peak resolution is often the limiting factor when attempting to identify a unit cell from powder data, so our technique gives a powerful advantage for cell determination.

Having said that, one could imagine a couple techniques for incorporating powder diffraction data. It's well known that full-profile refinement is excellent for obtaining accurate unit cell parameters. The XFEL-determined structure could be used as a starting point for Rietveld refinement, refining either (1) lattice parameters only or (2) lattice parameters and full structure. These would be valuable extensions of our technique but we felt they were outside the scope of this proof of concept.

We have updated the manuscript to clarify the advantage of making the powder patterns this way. A comparison with laboratory source PXRD is also included.

- The authors mention the energy spread of the FEL. Can they quantify the spread? Can it be shown as a histogram over energy?

Yes. This is a standard feature of SASE XFELs. It is important to consider but not essential to this manuscript, so we have provided a literature reference on the topic.

- The authors first used a 'maximum value' search for the Bragg peaks, but this produced no useable result. Why did they not use the dials.spot-finder from the beginning? It seems that the spots were normalised by their energy, i.e. converted to d-spacing. This seems an obvious way to go. Is this not so obvious?

The "maximum value" technique was a holdover from previous work in Brewster 2015. We appreciate the reviewer's perspective here and we agree the comparison of maximum values vs. spotfinder can be omitted.

- The authors cite N. K. Sauter (2015) for improving data intensities, but did not make use of this improvement. Why not?

The Sauter postrefinement method would become unstable for sparse smSFX frames, given that the number of observations on each image (3-10 intensities) is on the same order as the number of unknown parameters. We have removed this reference for clarity because the technique is not applicable here.

- Data were integrated to 1.15Å, but for refinement, data were cut to 1.2Å.

Why? This throws away more than 10% of the data.

The final resolution limits were set to give 10-15x multiplicity in the outer shell. This strict cutoff is required because all individual observations are partial. We clarify this in the Methods section.

- Similar to (1): The authors indicate a space group ambiguity between C2/c and Cc. In such cases, one would continue with both until a discrimination is clear, either because the structure can only be solved in one space group, or because refinement yields better results. This step should be included. Sometimes, of course, one cannot decide between space groups. Also in this case, data should be presented.

We extensively discuss the issue of thiorene space group assignment below, in the paragraph beginning "Regarding the phenyl disorder...". In short, we explored the possibility that the structure could be ordered in Cc instead of disordered in C2/c, but the refinement was not improved. We did find that thiorene gave much better results (R1 12%, improved from 19%) when modeled in Cc.

- What is the standard uncertainty in the cell parameters?

Unit cell refinement was not possible with our current methods; thus the cell parameters are taken directly from the SVD-Index cell search and it is not possible to assign a meaningful uncertainty. In macromolecular XFEL diffraction, we typically refine an independent unit cell for every indexed frame (which of course is also not the best way to assign an uncertainty on the cell parameters). In this small-molecule experiment, we cannot refine an independent cell for every frame because the data are so sparse. We have not yet implemented a working method for joint refinement of a single unit cell against a full data set, so at present we are leaving this as an improvement for future work. Incorporating powder data would be helpful here as discussed above. Finally we note that standard uncertainties on unit cells refined against single-crystal data are also famously problematic: <https://doi.org/10.1107/S010876810000269X>

Minor technicalities:

- The title should be changed from "De novo" to "Ab initio". "Ab initio" (cf. Int.

Tables Cryst. Vol F) in crystallography refers to phasing based only on intensities, which is the important point of this work.

We thank the reviewer for this comment and have updated the manuscript title accordingly to "*Ab initio* structures of microcrystalline hybrid materials by serial femtosecond crystallography" to reflect the correct term.

- The term 'mithrene' appears the first time on p. 3, line 75, but is used there like it has been introduced before (unless 'Summary' is considered part of the main text).

Mithrene, thiorene, and tethrene are now explained on first use.

- p. 3, line 79 "... shown in the supporting materials". A reference to Fig. S1, S2 should be added here.

This phrase is no longer in the manuscript.

- p. 4, around line 89: "results in few (3-10) reflections per frame": this is a repetition of p. 3, line 75.

This phrase is no longer used.

- The abstract describes mithrene to have 9 independent non-H atoms. In the context of the manuscript, this is exaggerated, as only 4 independent non-H atoms were refined, plus the shift and rotations of the benzene ring. However, with mild restraints (SADI for 1,2- and 1,3-distances of the C6-ring), all atoms can indeed be refined independently, which the authors may want to do.

The phrasing "9 independent non-H atoms" is no longer used.

Regarding a restrained refinement of the disordered phenyl ring, we appreciate the suggestion and we tried a few possibilities here. With (1) only SAME instructions or (2) only floating DFIX restraints on 1,2- and 1,3-distances, the phenyl geometries were recognizably distorted. We were able to get a reasonable restrained refinement using the following restraints:

same se3 c1a c6a < c2a

same se3 c1b > c6b

same se3 c1b c6b < c2b

dfix 41 .02 c1a c2a c2a c3a c3a c4a c4a c5a c5a c6a c6a c1a

dfix 41 .02 c1b c2b c2b c3b c3b c4b c4b c5b c5b c6b c6b c1b

dfix 51 .04 c1a c3a c2a c4a c3a c5a c4a c6a c5a c1a c6a c2a

dfix 51 .04 c1a c3a c2a c4a c3a c5a c4a c6a c5a c1a c6a c2a

flat se3 c1a > c6a

flat se3 c1b > c6b

For the restrained refinement there are 62 parameters and 77 restraints, compared to 38 parameters and 9 restraints in the constrained refinement; ultimately achieving identical results and agreement factors in both cases. We felt that building a minimally parametrized model without requiring a lot of restraints was the most consistent approach. With the prospect of higher energy XFEL diffraction (~20 keV expected this year at LCLS) we are hopeful that future refinements will more resemble typical single-crystal work.

- Fig. 4C: the comparison with the SXRD structure (Cuthbert et al) does not show very interesting details. One could instead use this panel to show the disordered benzene ring down the C-Se bond. This would be more informative.

In the present revision we feel the disordered phenyl group is not a central point, since the comparison of the inorganic skeletons and the resulting differences in photophysics are the more remarkable result. The comparison with the literature structure (now figs. 3C,D) is included to illustrate that the overall results are indistinguishable; thus we chose the orientation to match panels A and B to clarify the visual presentation.

Referee #3 (Remarks to the Author):

The manuscript presents a determination of the crystal structure of mithrene by means of serial femtosecond crystallography (SFX). The key claim of the manuscript is that it demonstrates the feasibility of determination of small-unit-cell materials by SFX.

The aim of this review is two-fold: to assess the technical quality of the structure determination and correctness of the claims in the manuscript, and to evaluate the novelty, originality and importance of the work.

Concerning the first point, I can say that, technically speaking, the goal of the manuscript was fulfilled and the demonstration that small crystal structures can be solved by SFX was successful. My more detailed comments on certain questionable aspects of the structure analysis follow at the end of the report.

It is more difficult to assess the aspects of novelty and importance. A manuscript can be groundbreaking either thanks to the novelty of the presented approach, by the importance of the results or by the (potential) impact of the presented procedures or methods. Unfortunately, as much as I regret it, I must say that I fail to see the groundbreaking nature of the presented manuscript in any of the three ways. Let me explain this opinion more specifically:

(1) novelty of the presented approach: the manuscript seems to be combining experimental and computational procedures known and presented before. The difference between "large molecule" and "small molecule" SFX is not fundamental, and "small-molecule" SFX just needs to overcome several specific computational difficulties, mostly related to the sparsity of individual diffraction patterns. Most of these difficulties, if not all, have been already solved elsewhere. The manuscript itself notes the work of Brewster et al. that used the same algorithms for the key indexing step. The manuscript, however, does not reference e.g. the work of Dejoie et al. (Serial snapshot crystallography for materials science with SwissFEL, IUCrJ 2, 361-370, 2015). Although this work mimicks the SFX data using standard synchrotron beam-line and the results are thus somewhat artificial, it addresses most of the problems (sparseness of the patterns, indexing, merging...). Another work addressing indexing of

sparse small-molecule frames is Dejoie et al., J. Appl. Cryst. (2020). 53, 824-836.

Both of the Dejoie et al articles were cited in the manuscript as submitted (refs 14, 15). We agree with the reviewer that this cited work simulates some aspects of the manuscript completed. We feel that this literature is an *ipso facto* justification of the effort we spent here to realize the methods experimentally.

Moreover, a lot has been done on serial electron crystallography of small (non-macromolecular) materials (e.g. <http://scripts.iucr.org/cgi-bin/paper?S1600576717005854>), and this work is also neither mentioned nor referenced in the present manuscript (see also below for more comments on electron crystallography).

We have considered serial electron crystallography in more detail in our revised manuscript, and will discuss further in the later comments.

(2) importance of the specific results: the structure of mitherene is known and it is not the ambition of the manuscript to present a new important structure. This aspect is thus also not relevant here.

We have remedied this critique by including two new structures solved by smSFX, tethrene and thiorene, in the updated manuscript. The structure of thiorene corrects a prior PXRD-determined structure of the Ag-S coordination environment and Ag-Ag lattice presented by Dance, et al (1991), which we believe is an important contribution to the field. Additionally, the structures of both tethrene and thiorene solve a long term question of the observed divergence in optoelectronic properties in the MOChA series.

(3) potential impact of the presented method: SFX is only one of several methods to solve structures from microcrystalline samples. Two other prominent examples are x-ray powder diffraction and electron single crystal diffraction. In the manuscript very little is dedicated to the comparison of the presented method to the "competition". It seems to me that the amount of material needed is relatively large and thus it should be possible to collect a standard powder x-ray diffraction (PXRD) pattern from it. At this size of the unit cell a solution from powder diffraction would most likely not pose big problems. Where PXRD fails is either for phase mixtures - but SFX suffers from the problem of phase mixtures as well - or larger unit cells, but there the SFX approach moves closer to the well established "large unit cell" SFX methods.

We tried and failed for many years to determine these structures by powder diffraction; furthermore the incorrect description of thiorene in Dance 1991 shows this was a difficult problem. Fundamentally, the problem of peak overlap is a major challenge for both unit cell determination and recovering intensities, even for a modestly sized unit cell as in this study. In SFX we naturally avoid the overlap problem at both stages by measuring one crystal at a time.

Even more important is the comparison to 3D electron crystallography methods (3D ED, aka MicroED). The manuscript deals with electron diffraction in a

single sentence, stating that "electron diffraction is not easily applied when samples are radiation sensitive or experimental conditions are near standard temperature and pressure". This is true only in the last aspect - electron diffraction cannot be performed on crystals at ambient pressure. Otherwise, numerous studies have been done on very radiation sensitive materials both at ambient and low temperature. Moreover, serial electron diffraction method (SerialED) has been developed and is in use, which overcomes most of the remaining difficulties with radiation sensitive materials. Thus, electron diffraction is a relatively (in comparison to SFX) easily accessible method solving most of problems small-molecule SFX would be attractive for. I thus see very little room for problems in small molecule crystallography that cannot not be tackled by other, more accessible and less involved methods, and would require the use of SFX techniques.

In summary, proving that SFX can be successfully used to solve a small-molecule structure is certainly interesting, it is opening new possibilities and enriching the landscape of available methods to deal with microcrystalline materials, but I fail to identify the necessary aspects that would make this development a significant breakthrough.

While we agree with the reviewer that Micro electron diffraction (MicroED) and serial electron diffraction (serialED) are useful for the structure determination of microcrystals, there are limitations to these methods as well. In the case of extremely radiation sensitive and desolvation sensitive materials like metal-organic frameworks and the class of materials studied in this manuscript, MOChAs, ED methods become no longer compatible due to the need for vacuum conditions. SerialED has also been successful in characterizing structures of materials in the zeolite class that generally display little-to-no radiation sensitivity. For some low-symmetry materials,

reference materials and zeolite-specific framework searches were used to aid scaling and structure solutions. All three representative MOCha samples have lower symmetry spacegroups and display extreme radiation sensitivity, and all three structure solutions were determined using *ab initio* methods.

For more general applications, SerialED methods can sample 500 crystals in an hour (Crystal, 2021), whereas smSFX has the capability to sample ~400,000 crystals in 2-4.5 hours, depending on the repetition rate of the XFEL source. This makes smSFX a much more robust crystal screening method by several orders of magnitude. The advent of MHz repetition rates will also make smSFX truly high-throughput, where compatible sample delivery conditions can provide complete datasets in minutes.

Getting back to the actual contents of the manuscript, here are a few more specific comments:

- line 43: the references to electron diffraction works are not very representative. If structure solution of small-molecule compounds by electron diffraction is referenced, then either the primary works by Kolb et al. should be referenced (possibly together with the more recent works by Jones et al. - reference 3, and Gruene et al. -

<https://onlinelibrary.wiley.com/doi/full/10.1002/anie.201811318>), or the big review by Gemmi et al., which summarizes most of the relevant literature before 2019: <https://pubs.acs.org/doi/abs/10.1021/acscentsci.9b00394>.

We thank the reviewer for the suggested microED works to cite and have added the representative citations into the manuscript accordingly.

Twinning: I am not convinced by the description and argumentation supporting the discussion about twinning. If there was a full indexing ambiguity, then the twin ratio would be expected to be close to 50:50. If the ambiguity would be present only for some orientations and not for other, then using an overall twin law would not be the appropriate way to describe the situation. In any case, when the preliminary model was used to re-index the data and remove the indexing ambiguity by correlation with the calculated intensities, then the twin ratio should gradually shift towards 100:0. However, in reality it remains constant and only in the last step changes slightly from 81:19 to 85:15. This does not make sense to me and makes me believe that there is another effect involved. Have you considered the possibility that individual microcrystals can be twinned and the data themselves come from twinned crystals? If this is the case, then the whole analysis would be complicated, because the

twin ratio for each crystallite may be different. See also my next comment.

See below for more on twinning.

Disorder in the structure: You found disorder in the phenyl ring, while the reference structure (Cuthbert et al.) does not report this disorder. This discrepancy is not even mentioned in the manuscript, and there is no discussion about the disorder. The disorder is not enforced by symmetry and it is thus also not obvious why the ratio of the two orientations should be 50:50. Have you tried to refine the ratio? The structure is layered and I thus suspect it could be prone to polytypism.

Cuthbert states: "disordered phenyl ring was modeled with 50:50 occupancy". We have clarified our manuscript by explicitly stating that the disorder is by rotation around the C-Se bond. We had hoped to better resolve this disorder with the SM-SFX experiment, however, our measurements (including refining the occupancies for the disorder) confirmed the earlier report.

The (possible) twinning points also to this direction. Could it be that the disorder is in reality not disorder, but rather an unaccounted-for twinning? Maybe the observed disorder and twinning are just two demonstrations of the same effects. I think that a more detailed analysis of these aspects is necessary.

We appreciate these insightful comments on twinning and disorder.

We agree it is surprising that the twinning fraction starts at 81:19 and only improves to 85:15. There are two effects at work here: First, the pseudo-orthorhombic metric giving rise to the indexing ambiguity is not perfect; the approximate oF supercell has lattice parameters $a=5.94$, $b=7.32$, $c=58.14$, $\alpha=\beta=\gamma=90$, $a\neq b\neq c$. Therefore the indexing is partially biased toward the correct sense. Second, since individual frames are sparse and all reflections are partial, the correlation coefficients determined for single shots are quite noisy and therefore the reindexing is not perfect. These two factors explain the surprising results. Refinement of a twinning fraction is an imperfect expedient, but it does improve the refinement.

These arguments are substantiated by the thiorene and tethrene data sets, where the metric is even further from pseudo-orthorhombic, but we still observe a small residual fraction (<5%) of misindexed frames that we model as twinning.

Regarding the phenyl disorder, the ratio of the two disordered positions does refine to very near 1:1, which is reasonably interpreted as a warning sign that a rotation in the assigned space group

has actually been simulated by twinning. In this case we note that a twofold rotation around b does appear to exchange the two disordered positions of the C6H5 group. Thus it's important to consider the possibility that the true structure is ordered in Cc . In our hands, refinements of the model ordered in Cc (with or without a Flack parameter) gave unsatisfactory results with severe instability and no improvement in the agreement factors.

It has been noted that light-atom disorder can be hard to distinguish from twinning (ref.: W. Hoenle and H.G. von Schnering, *Z. Krist.*, **184** (1988) 301-305; as cited in the ShelXL manual). This is an especially difficult case because (1) the disordered part is relatively small and contributes weakly to the scattering; and (2) the orientation of the putative polar axis in Cc is lost in the serial experiment. Thus we feel that the best practical refinement choice is a disordered model in $C2/c$. A high-quality single-crystal study with Cu radiation could resolve the issue, but owing to the difficulty of crystal growth for the x-rene family, that is not a trivial undertaking either.

Interestingly, thiorene did present a clear case of a refinement that was substantially improved by lowering the symmetry to Cc .

Reviewer Reports on the First Revision:

Referee #1 (Remarks to the Author):

The authors have expanded the number of structures determined using serial femtosecond crystallography to three and have thus proved the generality of the method. The results do represent a step change in the way that crystal structures of small molecules can be determined and this has enormous implications for the future.

The authors have addressed the majority of the issues raised by the referees and the manuscript is much stronger. I am convinced by the response that they have given to my criticisms. It is very exciting that the resolution of the data collected will improve to the sub-Angstrom level in 2022, which will really make the determination of high quality small molecule structures accessible. I would argue that there will remain some issues concerning the application of absorption corrections for small molecules with more than one heavy metal (third row transition metal for example) but this is really only a minor point.

I would recommend acceptance of the manuscript in its present form after technical editing, and if the other referees recommend publication in Nature (rather than Nature Communications) I would be happy to support this decision.

Referee #2 (Remarks to the Author):

Schriber et al present the revised version of their manuscript "Ab initio Structures of Microcrystalline Hybrid Materials by Serial Femtosecond Crystallography". The new version addresses all concerns of the reviewers.

It is very impressive that the authors could include two more structures, tethrene and thiorene. In addition to the impressive methodical work, these new structures help interpret the properties of the chemical compounds. This manuscript addresses a broad community, since the applications of (time-resolved) smSFX are huge and cover a great number of disciplines of science.

This manuscript includes the very important information that 20keV FELs will be available in the future. The authors should express this information (in the Conclusions section) in the words of crystallographers, like they do in response to reviewer 3: 0.68Å will be the resolution limit with this energy. With this information presented in the Conclusions (and possibly abstract), every scientist who takes benefit from crystallography will be intrigued by the prospect this work is opening. Also the updated figures (esp. Fig 4A-C) greatly improve the manuscript and make it even more attractive to the non-specialist scientist.

Referee #3 (Remarks to the Author):

I appreciate the improvements made by authors to the manuscript. They address most of my comments and visibly improve the quality of the manuscript.

I still cannot agree with the account given by the authors on electron diffraction. I understand the motivation of the authors to showcase the weaknesses and somewhat suppress the strengths, but the statements made about 3D ED (this term should be preferred over MicroED, which is a name for just one particular flavor of the method) are not accurate and are biased. 3D ED methods have been used to solve hundreds of structures of sometimes very unstable materials, *ab initio*, without any help from modeling or external sources. Serial ED was also used on unstable materials including proteins, not only on "zeolite materials that generally display little-to-no radiation sensitivity" as stated in the manuscript, see e.g. the works of Robert Bücker. Sensitivity to vacuum can be easily circumvented by measurement at cryo conditions, which is in vast majority of cases not a problem. Contrary to SFX, the samples for 3D ED can be prepared dry, no need for a medium to disperse the crystals in. Talking about the specific case presented in this manuscript, I have little doubt that the structures of thiorene, mitherene and tetrene could be easily solved by 3D ED methods.

However, it is not the point to argue, which of the methods is better. My point is only that the alternative methods should not be displayed in a biased and dismissive manner.

In my view (which again might be biased), a fair comparison of the two methods is the following:
SFX:

pros:

- + no need for cryocooling
- + possibility to analyze quickly a very large number of crystals

contras:

- very expensive and not easily accessible method
- necessity to use dispersive medium and complicated sample delivery system(? not sure on this point, actually)
- relatively large amount of sample needed (compared to 3DED)
- relatively slow, at least at present stage

3E ED:

pros:

- + can be performed really on one single crystal
- + easily accessible, comparably cheap
- + can analyze much smaller crystals than SFX (useful data from single crystals down to X0 nm)
- + fast (one dataset sufficient for structure solution is collected in at most 5 minutes, often just

about one minute).

contras:

- for beam sensitive materials it requires cryoconditions
- SerialED throughput not as fast as SFX - typically hundreds to a few thousands snapshots per hour.
- exhibits more indexing ambiguities than SFX due to the flatness of the Ewald sphere (although this is a minor problem in practice)

Despite of the differences, my personal opinion is that >90% of problems that cannot be solved by classical single-crystal x-ray diffraction and do not require special sample environment like high pressure, can be solved either BOTH by SFX and 3D ED or by none of them.

As I indicated in my previous review, I do not have any severe objections to the technical aspect of the manuscript and I think it can be, after the modifications made by the authors in the revision, published. I believe the journal editor is now best positioned to make the decision on the importance and significance of the proposed method.

Author Rebuttals to First Revision:

Referee #1 (Remarks to the Author):

The authors have expanded the number of structures determined using serial femtosecond crystallography to three and have thus proved the generality of the method. The results do represent a step change in the way that crystal structures of small molecules can be determined and this has enormous implications for the future.

The authors have addressed the majority of the issues raised by the referees and the manuscript is much stronger. I am convinced by the response that they have given to my criticisms. It is very exciting that the resolution of the data collected will improve to the sub-Angstrom level in 2022, which will really make the determination of high quality small molecule structures accessible. I would argue that there will remain some issues concerning the application of absorption corrections for small molecules with more than one heavy metal (third row transition metal for example) but this is really only a minor point.

I would recommend acceptance of the manuscript in its present form after technical editing, and if the other referees recommend publication in Nature (rather than Nature Communications) I would be happy to support this decision.

We appreciate the reviewer's comments on our revised manuscript. We agree that improvements like absorption correction will be important as the scope of applications increases, although we note that higher energy XFEL radiation will help reduce the size of the absorption problem.

Referee #2 (Remarks to the Author):

Schriber et al present the revised version of their manuscript "Ab initio Structures of Microcrystalline Hybrid Materials by Serial Femtosecond Crystallography". The new version addresses all concerns of the reviewers.

It is very impressive that the authors could include two more structures, tethrene and thiorene. In addition to the impressive methodical work, these new structures help interpret the properties of the chemical compounds. This manuscript addresses a broad community, since the applications of (time-resolved) smSFX are huge and cover a great number of disciplines of science.

This manuscript includes the very important information that 20keV FELs will be available in the future. The authors should express this information (in the Conclusions section) in the words of crystallographers, like they do in response to reviewer 3: 0.68Å will be the resolution limit with this energy. With this information presented in the Conclusions (and possibly abstract), every scientist who takes benefit from crystallography will be intrigued by the prospect this work is opening. Also the updated figures (esp. Fig 4A-C) greatly improve the manuscript and make it even more attractive to the non-specialist scientist.

We thank the reviewer for their response and appreciate their critique of the resubmitted manuscript. We have amended the manuscript to better express the XFEL energy upgrade in the conclusion. See lines 219-220. "Access to higher beam energies, with 25 keV expected at LCLS in 2022, will make <0.7 Å resolution available "

Referee #3 (Remarks to the Author):

I appreciate the improvements made by authors to the manuscript. They address most of my comments and visibly improve the quality of the manuscript.

I still cannot agree with the account given by the authors on electron diffraction. I understand the motivation of the authors to showcase the weaknesses and somewhat suppress the strengths, but the

statements made about 3D ED (this term should be preferred over MicroED, which is a name for just one particular flavor of the method) are not accurate and are biased. 3D ED methods have been used to solve hundreds of structures of sometimes very unstable materials, ab initio, without any help from modeling or external sources. Serial ED was also used on unstable materials including proteins, not only on "zeolite materials that generally display little-to-no radiation sensitivity" as stated in the manuscript, see e.g. the works of Robert Bücker. Sensitivity to vacuum can be easily circumvented by measurement at cryo conditions, which is in vast majority of cases not a problem. Contrary to SFX, the samples for 3D ED can be prepared dry, no need for a medium to disperse the crystals

in. Talking about the specific case presented in this manuscript, I have little doubt that the structures of thiorene, mitherene and tetrane could be easily solved by 3D ED methods.

However, it is not the point to argue, which of the methods is better. My point is only that the alternative methods should not be displayed in a biased and dismissive manner.

In my view (which again might be biased), a fair comparison of the two methods is the following:

SFX:

pros:

- + no need for cryocooling
- + possibility to analyze quickly a very large number of crystals

contras:

- very expensive and not easily accessible method
- necessity to use dispersive medium and complicated sample delivery system(? not sure on this point, actually)
- relatively large amount of sample needed (compared to 3DED)
- relatively slow, at least at present stage

3E ED:

pros:

- + can be performed really on one single crystal
- + easily accessible, comparably cheap
- + can analyze much smaller crystals than SFX (useful data from single crystals down to X0 nm)
- + fast (one dataset sufficient for structure solution is collected in at most 5 minutes, often just about one minute).

contras:

- for beam sensitive materials it requires cryoconditions

- SerialED throughput not as fast as SFX - typically hundreds to a few thousands snapshots per hour.
- exhibits more indexing ambiguities than SFX due to the flatness of the Ewald sphere (although this is a minor problem in practice)

Despite of the differences, my personal opinion is that >90% of problems that cannot be solved by classical single-crystal x-ray diffraction and do not require special sample environment like high pressure, can be solved either BOTH by SFX and 3D ED or by none of them.

As I indicated in my previous review, I do not have any severe objections to the technical aspect of the manuscript and I think it can be, after the modifications made by the authors in the revision, published. I believe the journal editor is now best positioned to make the decision on the importance and significance of the proposed method.

We appreciate these detailed comments. We agree that the discussion should not be framed as a competition between different methods for analyzing microcrystals, and we thank the reviewer for helping us improve the manuscript with these comments. We have revised the manuscript to emphasize that smSFX is a complementary addition to the existing advanced diffraction techniques. In the case of these samples, smSFX was the technique that ultimately worked.

We'd like to comment on some specific points:

-While XFEL beam time is rare, we hope that further improvements in instrumentation, reasonably expected to reduce data collection times by at least 25x in the next few years, will allow smSFX as a mail-in service as is currently practiced at some synchrotron beamlines.

-The sample delivery required some fine-tuning, but since the sample is simply a slurry in some solvent, we anticipate that our success here will be transferable to a wide range of additional samples.

-Another important benefit is the intrinsic time resolution of SFX, already used for mechanistic studies in many macromolecular systems. The same technique might be applied to materials systems, but we elected not to include discussion on this topic as it is beyond the scope of our present work.

On a few additional details:

-We have added an additional sentence introducing smSFX, where we acknowledge that the method we use for sample delivery, a liquid jet with solvated samples, can be sample consumptive compared to 3D-ED methods. See lines 86-88. "This can be sample consumptive, but there are adaptations available to conserve sample and even recirculate the suspension through the interaction point."

-We have amended the manuscript to correctly name the method as 3D-ED, as opposed to microED, per the reviewer's comment.